# Position-dependent function of human sequence-specific transcription factors

Sascha H. Duttke[1 ✉], Carlos Guzman[2], Max Chang[2], Nathaniel P. Delos Santos[2], Bayley R. McDonald[1], Jialei Xie[3], Aaron F. Carlin[3], Sven Heinz[2 ✉] & Christopher Benner[2 ✉]

Patterns of transcriptional activity are encoded in our genome through regulatory elements such as promoters or enhancers that, paradoxically, contain similar assortments of sequence-specific transcription factor (TF) binding sites[1–3]. Knowledge of how these sequence motifs encode multiple, often overlapping, gene expression programs is central to understanding gene regulation and how mutations in non-coding DNA manifest in disease[4,5]. Here, by studying gene regulation from the perspective of individual transcription start sites (TSSs), using natural genetic variation, perturbation of endogenous TF protein levels and massively parallel analysis of natural and synthetic regulatory elements, we show that the effect of TF binding on transcription initiation is position dependent. Analysing TF-binding-site occurrences relative to the TSS, we identified several motifs with highly preferential positioning. We show that these patterns are a combination of a TF's distinct functional profiles—many TFs, including canonical activators such as NRF1, NFY and Sp1, activate or repress transcription initiation depending on their precise position relative to the TSS. As such, TFs and their spacing collectively guide the site and frequency of transcription initiation. More broadly, these findings reveal how similar assortments of TF binding sites can generate distinct gene regulatory outcomes depending on their spatial configuration and how DNA sequence polymorphisms may contribute to transcription variation and disease and underscore a critical role for TSS data in decoding the regulatory information of our genome.

Each cell of an organism interprets the same genome in a unique way. At the heart of this process are sequence-specific TFs that orchestrate regulatory programs and interpret the regulatory sequence grammar inscribed in the genome[6–8]. How these regulatory programs are encoded is still largely enigmatic. Many regulatory elements contain sequence motifs for similar sets of TFs and most TFs display widespread binding to regulatory sequences, with variable and sometimes minimal consequences for gene regulation[9–11]. Consequently, we are largely unable to predict gene expression patterns from DNA sequence alone[12,13] and it is unclear how the transcription of most human genes is regulated. Previous studies have shown that TF-binding-site spacing, orientation and copy number, and affinity of TF binding sites can influence transcriptional output[5,6,14–16]. However, few generalizable rules exist for how TF binding sites construct gene regulatory programs, restricting our ability to rationally interpret our genome or understand how mutations in regulatory sequences impact gene regulation or manifest in disease[17].

## Preferential spacing of TFs relative to the TSS

The TSS is a landmark of gene expression, where regulatory signals are ultimately integrated to start transcription. Regulatory elements including promoters and enhancers as defined by active transcription initiation (hereafter collectively referred to as transcription start regions (TSRs))[18] often start transcription from several different TSS locations rather than a single site[19,20]. Capturing TSSs across different cell types or in response to stimuli revealed that TSS selection within TSRs can be highly dynamic[21,22]. We therefore set out to investigate motif grammar from the perspective of each individual TSS, rather than from the perspective of open chromatin, a specific protein or epigenetic state.

To do so, we developed HOMER2 (Methods and Extended Data Fig. 1), a suite of analysis tools to study DNA sequence motif enrichments accounting for both GC content and position-dependent nucleotide biases, for example, as found near TSSs (Fig. 1a). We next profiled human U2OS cell TSSs using capped small RNA sequencing (csRNA-seq)[21], which accurately captures the TSSs of both stable and unstable RNAs. Using these TSSs as anchors for de novo motif analysis[23] revealed that the binding sites of the most-enriched TFs had preferential localizations relative to active TSSs (Fig. 1b and Extended Data Fig. 2a). This preferential positioning was particularly apparent for sequences bound by ubiquitously expressed canonical activators such as NRF1, NFY, Sp1 and ETS-family TFs[6,24–27] (Extended Data Fig. 2b,c). In general, these activator binding sites were enriched upstream of the core promoter region

[1]School of Molecular Biosciences, College of Veterinary Medicine, Washington State University, Pullman, WA, USA. [2]Department of Medicine, Division of Endocrinology, U.C. San Diego School of Medicine, La Jolla, CA, USA. [3]Department of Pathology and Medicine, U.C. San Diego School of Medicine, La Jolla, CA, USA. ✉e-mail: sascha.duttke@wsu.edu; sheinz@health.ucsd.edu; cbenner@health.ucsd.edu

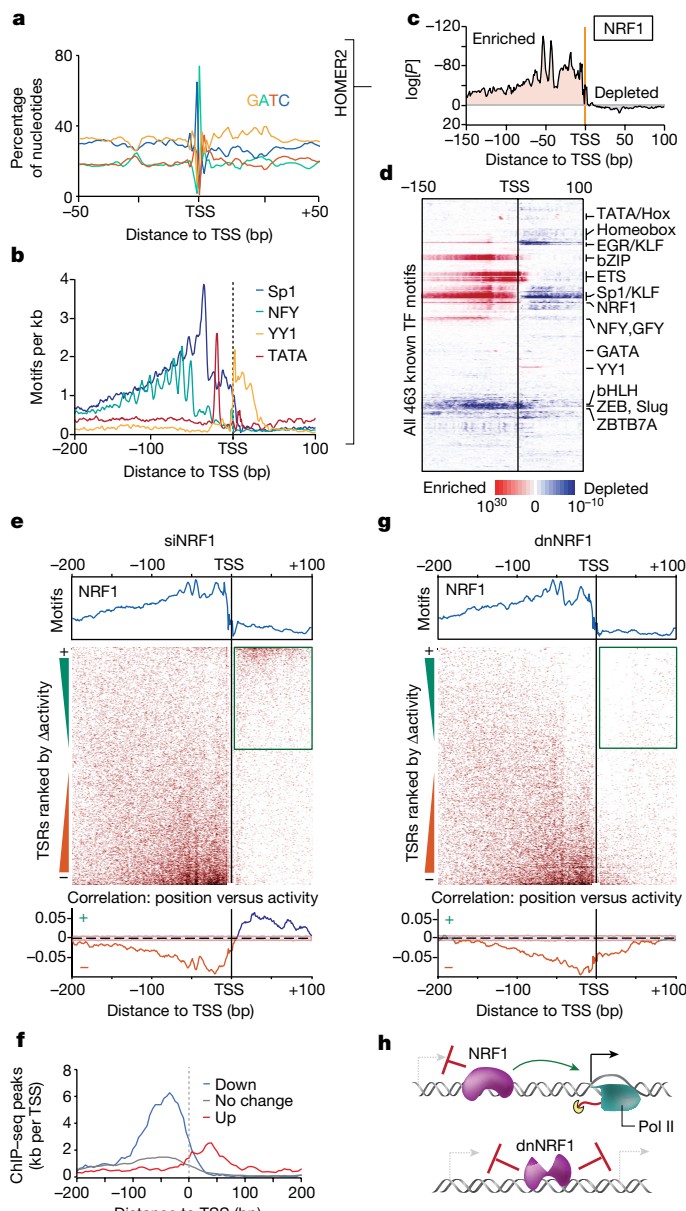

**Fig. 1 | TF function is position dependent. a**, Nucleotide frequency bias near TSSs of human U2OS cells. **b**, Many TF binding sites are enriched at specific positions relative to active TSSs genome-wide (TSSs). **c**, Integrating positional information and nucleotide biases using HOMER2 identifies TF binding site enrichment or depletion relative to the TSS, exemplified by NRF1. Statistical analysis was performed using two-sided Fisher's exact tests with Benjamini–Hochberg correction. **d**, Most TF binding sites are enriched in preferred positions. Enrichment or depletion of all 463 known human TF binding sites in the HOMER2 motif database relative to the TSS. A detailed version of this figure is provided in Supplementary Fig. 1. **e**, NRF1 function is dependent on the location of its binding site relative to the TSS. TSSs are ranked on the basis of their $\log_2$-transformed fold change in activity after *NRF1* knockdown (from gain to loss of transcription initiation, $n = 136,757$). TSSs with NRF1 binding sites (heat map, dark red) within their preferred localization pattern (blue graph; top) are repressed, while those with NRF1 binding sites downstream of the TSS were correlated with activation (or derepression; bottom). Analysis was performed using MEPP (Methods). **f**, TSSs downregulated after *NRF1* knockdown (siNRF1) display TF binding within its preferred upstream region while derepressed TSSs display NRF1 binding downstream, as assessed by anti-NRF1 ChIP–seq. **g**, NRF1 probably represses upstream TSSs through steric hindrance. Analysis of TSSs ($n = 136,344$) ranked from gain to loss of transcription initiation activity after overexpression of a transcription activation domain (TAD)-deleted dnNRF1 mutant shows repression when the NRF1 binding site (dark red) is located either upstream or downstream of the TSS, suggesting that TSSs found upstream of NRF1 sites are activated only after removal of the TF from the downstream DNA. **h**, Model for NRF1 TF function and NRF1-dependent TSS derepression.

promoter element[36] (Extended Data Fig. 2i–k). Consistent with the eminence of these elements in anchoring down the RNA polymerase and guiding TSS selection[20,36], this finding suggests positionally enriched TF binding sites may themselves direct TSS selection but that their impact is superseded by core promoter elements. Together, these findings highlight that many TF binding sites are enriched or depleted at specific positions relative to the TSS.

## TF position governs regulatory impact

We speculated that the preferred localization patterns of diverse TF binding sites are a reflection of TF function and may represent a superposition of the multiple mechanistic roles TF can have in regulating transcription initiation. To assess how the position of a TF relative to the TSS may affect its function, we knocked down *NRF1* and *YY1* in human U2OS cells using short interfering RNAs (siRNAs) (Extended Data Fig. 3a) and captured changes in transcription initiation 24 h later using csRNA-seq. We selected these TFs because their binding sites have strong positional preferences, and they are the only proteins known to bind to their respective motifs. Furthermore, NRF1 sites are preferentially located upstream of the TSS (Fig. 1c), while YY1 sites are unique for their preferred location downstream of the TSS (Fig. 1b,d). Consistent with previous reports that both regulators are potent activators[37,38], knockdown of *NRF1* and *YY1* diminished the csRNA-seq signal at 3,791 and 1,621 TSSs (<−1.5-fold, adjusted $P$ ($P_{adj}$) < 0.05), respectively. Moreover, knockdown of these strong activators also increased initiation at a comparable number of TSSs (3,971 and 1,160, respectively, >1.5-fold, $P_{adj}$ > 0.05). Follow-up transcriptome analysis showed that these ectopic TSSs not only resulted in alternative 5′ untranslated regions (UTRs) that, at times, produced new splice isoforms or open reading frames but also altered gene expression levels (Extended Data Fig. 4). TSSs with decreased or increased activity were frequently found within the same TSR, implying shifts in TSS selection within regulatory elements that may activate specific regulatory programs or help to buffer changes in gene expression. These findings mirror the results observed after depletion of *NFY*[39], and help to explain the observations that many TF binding events appear to be uncoupled from expected changes in gene expression[10,40,41].

(−40 to +40 bp, relative to the TSS)[28] and depleted near the active TSS, especially downstream, where the RNA polymerase II initiation complex is postulated to initially contact the TSR[29,30] (Fig. 1c,d and Extended Data Fig. 2c). One exception to this rule was YY1, a TF known for its dual role as a transcriptional activator and repressor[31,32]. The binding sites of repressors such as ZBTB7A/LRF[33] were depleted near active TSSs (Fig. 1d and Extended Data Fig. 2d). Although some TF binding sites were specifically enriched in distinct regulatory element types, such as bZIP TFs (such as AP-1) at enhancers[23,34], TF-binding-site-specific positional preferences were highly similar near TSSs for different transcript types (Supplementary Fig. 1). Enrichment patterns for several TF binding sites exhibited an approximately 10 bp periodicity, suggesting that the rotational position of TFs relative to the TSS affects transcription initiation[5,14,35] (Extended Data Fig. 2e). Positional preferences were conserved across cell lines, vertebrate species and TSS-detection methods and, in some cases, were restricted to genomic locations with cell-type-specific activity (for example, HNF1; Extended Data Fig. 2f–h). Positional preferences were less apparent for TF binding sites associated with weak transcription such as CTCF (Extended Data Fig. 2b,c). Moreover, spatial TF binding site enrichment patterns were more pronounced in the absence of the canonical initiator core

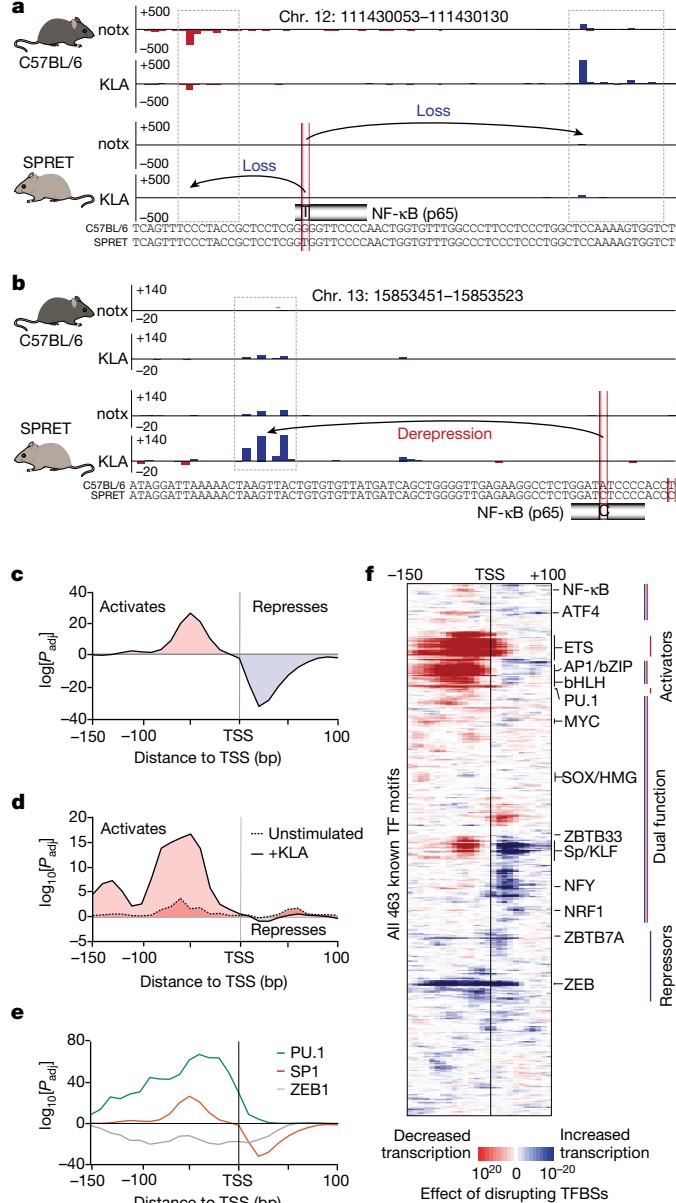

**Fig. 2 | Natural genetic variation reveals position-dependent function and distinct classes of TFs. a**,**b**, Natural DNA polymorphisms can have a major impact on gene regulation. Example loci where genetic variants eliminated NF-κB (p65) binding sites in the SPRET mouse strain (versus C57BL/6) and associated with either a reduction in downstream transcription initiation (**a**) or a derepression of transcription initiation when the mutated binding site was located downstream of the TSSs (**b**). Transcription initiation was measured in untreated (notx) or KLA-stimulated BMDMs from SPRET and C57BL/6 reference mouse strains (average of $n = 2$ biological replicates). **c**, The influence that variants in TF binding sites has on transcription initiation is dependent on their position relative to the TSS. Analysis of the genome-wide significance of the association between mutations in the Sp1 binding site (GC-box) and the change in transcription initiation, calculated for Sp1 sites as a function of their relative distance to the TSS. Positive log[$P_{adj}$] values indicate that mutations predicted to cause reduced Sp1 binding are more strongly associated with reduced initiation, whereas negative log[$P_{adj}$] values indicate that the mutated binding sites are more strongly associated with increased initiation (30 bp windows evaluated at 10 bp increments). Statistical analysis was performed using two-sided Mann–Whitney $U$-tests with Benjamini–Hochberg correction. **d**, Similar to **c**, but showing that mutations in the NF-κB (p65) binding sites exhibit stronger positional associations after 1 h of KLA stimulation (dotted versus solid line). **e**, The functional impact of mutating TF binding sites (TFBSs) generally follows one of three distinct patterns: pure transcriptional activators (PU.1), pure transcriptional repressors (ZEB2) and dual-function TFs (Sp1) that can activate or repress transcription initiation in a position-dependent manner. **f**, Position-dependent activity was evaluated for mutations impacting 463 known human TF motifs (not all are expressed in BMDMs). A detailed map of this figure is provided in Supplementary Fig. 2.

Similarly, *YY1* depletion resulted in the downregulation of TSSs with YY1 binding sites within its preferential zone immediately downstream of the TSS, whereas upregulated TSSs typically had a YY1 site further downstream (Extended Data Fig. 3f–h). Position-specific effects mirroring the TF binding site's natural enrichment were also identified when reanalysing published *NFY* knockdown data from mouse embryonic fibroblasts[39] (Extended Data Fig. 3i). Together, these results indicate that NRF1, YY1 and NFY can activate but also directly inhibit transcription initiation, depending on the position of their binding relative to the regulated TSS (Fig. 1h).

## Suppression of TSS by steric hindrance

To further examine the mechanisms underlying TSS activation after activator TF knockdown, we ectopically expressed a dominant-negative NRF1 (dnNRF1) mutant[45]. In contrast to siRNA knockdown, overexpression of transactivation domain-deficient NRF1 resulted in the downregulation of all TSSs near dnNRF1-bound sites, even those located downstream of the TSS (Fig. 1g and Extended Data Fig. 5a–d). TSSs activated after *NRF1* knockdown therefore probably depend on the clearance of the TF from the DNA, substantiating a model in which preferred spacing among TFs and the RNA polymerase II complex is critical for effective transcription initiation. Binding outside of these preferred positions inhibits RNA polymerase II recruitment and/or initial elongation, probably by steric hindrance (Fig. 1h), similar to the function of canonical prokaryotic repressors[46], Rap1 in yeast or CTCF in vertebrates[47,48]. These findings highlight the importance of accurate TSS positional information to decode TF function, and provide an explanation as to why it has been so challenging to predict gene expression programs from DNA sequence alone[12,13].

## TF function revealed by genetic variation

Genetic variation that naturally occurs between individuals offers an opportunity to study the functional impact of genetic variation on gene regulation[49,50]. As many of these variants affect TF binding sites, we captured the TSS landscape of bone-marrow-derived macrophages

Notably, ranking TSSs by their change in initiation frequency after TF knockdown revealed a clear bimodal distribution of the cognate TF binding sites: TSSs that were downregulated after *NRF1* knockdown were enriched for NRF1 binding sites upstream of the TSS, where the motif is preferentially located relative to active TSS locations genome-wide. By contrast, TSSs with increased transcription after *NRF1* knockdown had NRF1 binding sites positioned downstream of the TSS, where the motif is naturally depleted (Fig. 1c,e). Integrating ChIP–seq validated that downregulated TSSs had NRF1 bound predominantly upstream, whereas TSSs that were activated after *NRF1* knockdown had NRF1 bound downstream of impacted TSSs (Fig. 1f). Together, these findings demonstrate that NRF1 can both activate and inhibit transcription initiation, depending on its location relative to a TSS. The function of NRF1 is therefore position dependent. Given that the majority of strongly regulated sites contain an NRF1 binding site that, as assessed using chromatin immunoprecipitation followed by sequencing (ChIP–seq), is commonly bound by the TF (Fig. 1f and Extended Data Fig. 3b–e), the observed NRF1-dependent inhibition of transcription initiation in *cis* is probably distinct from previous reports showing that the loss of an activator can lead to activation through secondary effects in *trans*[10,40,42–44].

(BMDMs) from two mouse strains (C57BL/6 versus SPRET) to assess how sequence variants in TF binding sites impact transcription initiation as a function of their position relative to the TSS (Fig. 2). A key advantage of exploiting natural genetic variation is that it can be used to unbiasedly assess the impact of all TF binding sites on transcription, including those where TF redundancy may hinder analysis on the protein level under natural, unperturbed conditions.

To resolve distance-dependent functions of TFs, we used HOMER2 and its ability to normalize for positional single-nucleotide variant biases (Extended Data Fig. 5e,f) to assess the relationship between genetic alterations in TF binding sites and regulatory phenotypes. Contrasting 42.9 million single-nucleotide polymorphisms (SNPs) and 80,988 differentially regulated TSSs ($P_{adj}$ < 0.25) between the mouse strains revealed that the influence of genetic variants disrupting TF binding sites on initiation levels is position dependent (Fig. 2c–f). For example, variants that weaken consensus binding sites of NRF1 or NFY at their preferential positions relative to the TSS were strongly associated with reduced transcription (Extended Data Fig. 5g,h), consistent with the steric requirements of transcription complex assembly[29,30].

Analysis of all motifs in the HOMER2 database revealed that sequence variation found in 300 of 463 motifs is significantly associated with distance-dependent changes in transcription initiation ($P_{adj}$ < 0.01 in at least one position). Clustering of TF-binding-site patterns stratified three major classes: pure transcriptional activators, pure transcriptional repressors and dual-function TFs (Fig. 2e,f). For dual-function TFs, the location of the binding site relative to the TSS determined their role in activating or repressing transcription, usually segregated by their localization upstream or downstream of the TSS, respectively. This group includes the binding sites of the strong, ubiquitous TFs that are typically characterized as activators (Sp1/KLF, NRF1 and NFY), MYC, RUNX and other TFs that physically interact with the RNA polymerase II complex (Fig. 2f). These results were also verified using an alternative analysis approach[51] (Supplementary Fig. 2). The function of TFs that bind to these sites is therefore highly dependent on their relative position to the TSS. Pure activators, exemplified by the lineage-determining TF PU.1, included TFs of which variants with weakened consensus binding sites are generally associated with a reduction in transcription. These findings are consistent with previous reports suggesting that the TFs binding to these sites promote transcription through indirect mechanisms, including chromatin opening and epigenetic modification[52,53]. The third cluster of pure repressors encompassed TF binding sites that were associated with transcription activation when disrupted, and included the well-characterized repressors ZEB2[54] and ZBTB7A (also known as LRF)[33].

To assess whether position-dependent TF function extends to signalling-dependent TFs, we stimulated macrophages from both mouse strains with the TLR4 agonist Kdo$_2$-lipid A (KLA), which elicits a strong innate immune response[55]. Consistently, this stimulation led to a much more prominent signature for the binding sites of the main response factor, NF-κB[56] (Fig. 2d, Extended Data Fig. 5i,j and Supplementary Fig. 2). These data highlight position-dependent activity of TFs as a widespread phenomenon that extends to signalling-dependent TFs and argue that genome-wide analysis of TF binding sites and their spatial constraints relative to the TSS provides unbiased insights into TF function, thereby providing a tool to bridge the gap between DNA sequence and transcription.

### TSS-MPRA confirms TF positional function

To directly assess how the position of TF binding sites within regulatory sequences affects transcription en masse, we developed a massively parallel reporter assay (MPRA) strategy to capture transcription initiation at base resolution[57] (TSS-MPRA; Fig. 3a and Methods). As exemplified for a TSR found at the *EIF2S1* locus, TSS-MPRA accurately captured the relative initiation frequencies and locations of TSSs observed in vivo by csRNA-seq (Fig. 3b and Extended Data Fig. 6a).

To examine position-dependent TF function, we next synthetically inserted the binding site of six TFs (Sp1, NRF1, NFY, YY1, p53 and CTCF) or a control sequence predicted to not be bound by any known TF at positions −50 bp, −20 bp or +25 bp relative to the TSS into 13 different TSRs (Supplementary Table 2) and performed TSS-MPRA. As an internal control, and to mitigate potential barcode-dependent biases, each construct was paired with four distinct barcode sequences (Methods). In agreement with our results above, TF binding site insertion at −50 bp or −20 bp relative to the TSS stimulated initiation approximately fourfold for Sp1 and NRF1, and twofold for NFY (Fig. 3c). By contrast, insertion of the same sites at +25 bp reduced initiation by twofold to fivefold. Insertion of a CTCF site at the +25 position was even more repressive, consistent with the strong DNA binding and insulator function of CTCF[48,58]. By contrast, binding sites for YY1, which evolved to function at or just downstream of the TSS[59], showed the inverse trend, with significant increases in transcription when positioned at +25.

Binding of a TF at positions −20 bp or +25 bp relative to the TSS is generally thought to be sterically incompatible with RNA polymerase II complex assembly[60]. The negative regulatory phenotype observed for most TF binding sites when located at +25 bp, but less so when upstream of the TSS at −20 bp, proposes that the downstream core promoter region is critical for initial TFIID DNA recognition, while TFs bound to the upstream promoter region may be more readily displaced after TFIID has docked. These findings lend functional support to the cryo-electron-microscopy-based model that posits that TFIID initially interacts with TSRs in the downstream promoter region before transitioning to the upstream core promoter region[29,30,60].

A notable exception was the tumour suppressor p53. The p53 binding site lacks natural preferential spacing (Extended Data Fig. 5k) and activated transcription across all tested positions, resembling the behaviour of pure activator TFs uncovered in our natural genetic variation analysis. This finding is consistent with previous reports[44,61,62] and may provide an explanation for the unique potency of p53 to rewire transcription networks by itself. Together, these results corroborate the position-specific impact of TF binding sites on transcription initiation and suggest that there are different classes of TFs with distinct position-specific effect profiles.

### TF interactions influence TSS use

Synthetic recreation challenges our true understanding of observed biological phenomena. We therefore designed a synthetic 150 bp DNA sequence lacking known TF binding sites and generated thousands of variations thereof by placing a binding site for NRF1, NFY, YY1 or Sp1 in 2 bp increments across the entire sequence. Capturing both initiation strength and positions using TSS-MPRA revealed that insertions of TF binding sites frequently resulted in de novo TSSs (Fig. 4a, Extended Data Fig. 6b,c and Extended Data Fig. 7a–e), further supporting our observation that the TFs binding to these sites can strongly influence the recruitment and positioning of RNA polymerase II. Aggregating the sites of transcription initiation for each construct as a function of the distance between the TF binding site and the TSS (Fig. 4b,c and Extended Data Fig. 7f) revealed a position-dependent activity pattern resembling the binding site's natural enrichment relative to active TSSs, similar to our previous results (Fig. 1d and Extended Data Fig. 2c). Thus, multiple independent experimental and analytical approaches revealed an extremely similar pattern for a given TF (Fig. 4d and Extended Data Fig. 8a–d). While many activator TFs shared an overall preference for binding the −50 bp region relative to the TSS, similar to a fingerprint, each TF exhibited a unique pattern. These findings propose a model in which human TFs can directly guide transcription initiation based on the consensus of their unique spatial–functional profiles (Extended Data Fig. 8e).

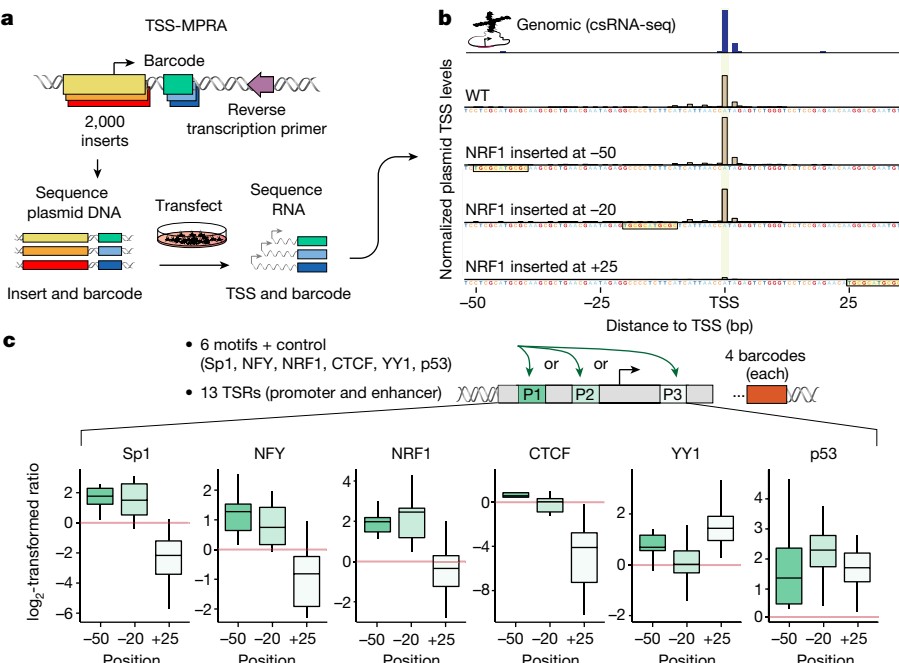

**Fig. 3 | Capturing the TSSs of thousands of TSR variants confirms the position-dependent function of TFs. a**, Schematic of TSS-MPRA, which accurately captures the TSSs and activity from thousands of plasmid cloned DNA sequences. **b**, Example of normalized TSS-MPRA transcription initiation data from four inserts designed based on a TSR in *EIF2S1* promoter (from −110 bp to +42 bp relative to the primary TSS). Three of the inserts show the impact of placing the NRF1 binding site at different positions relative to the TSS. The in vivo genomic TSS levels, as measured using csRNA-seq analysis in HEK293T cells, is shown at the top. WT, wild type. **c**, Synthetic TF binding site insertions confirm the position-dependent function of TFs. Summary of TSS-MPRA data for six TF binding sites inserted at three positions relative to the TSS. The impact of inserting the TF binding site was measured as the log ratio of normalized transcript levels versus the wild-type control. $n = 13$ distinct promoters, enhancers and other TSRs, and each insert was redundantly encoded with 4 different barcode sequences. The box plots show the median (centre line), 25th and 75th percentiles (box limits), and the minimum and maximum values (whiskers) for each position.

To test this model, we repeated the NRF1 sweep through a natural enhancer region adjacent to the *TOB2* gene. The activity pattern obtained by TSS-MPRA initially resembled that of the NRF1-binding-site sweep through our synthetic motif-depleted sequence, but then diverged with a more pronounced 10 bp helical periodicity upstream of −50 bp relative to the TSS (Fig. 4e). We hypothesized that these differences in the spatial activation profiles could be due to the presence of other TF binding sites in the enhancer region, such as the NRF1 site naturally found at −50 bp (Extended Data Fig. 8f). Indeed, repeating the NRF1-binding-site sweep in the *TOB2* enhancer in which the innate NRF1 site was mutated revealed a pattern that now closely resembled the synthetic NRF1 sweep (Fig. 4e). These results highlight that transcription initiation is affected by the spatial relationships not only between TF binding sites and the TSS, but also between the TF binding sites themselves.

To further examine the relationship between TF binding site spacing and transcription initiation, we selected all TSRs active in U2OS cells that contained at least two activator TF binding sites of interest within 300 bp of one another and the primary TSS, such as Sp1 and NRF1 motifs. Sorting these TSRs based on the distance between the sites and visualizing transcription initiation patterns measured by csRNA-seq revealed several trends that are consistent with the position-dependent functions of TF binding site pairs in vivo. First, TSSs were predominantly located at the preferred position characteristic for the 3′ TF in each TF binding site pair (Fig. 4f ('2' regions)). Second, TSSs were depleted in between the two TF binding sites, consistent with the ability of these TFs to suppress initiation when binding downstream of the TSS, indicating that the most-3′-binding TF has a dominant role in activating TSS selection (Fig. 4f ('1' regions) and Extended Data Fig. 9).

To gain further insights into TF-mediated TSS selection, we reanalysed our own as well as published TF knockdown data for *NRF1, YY1* and *NFY*[39] with a focus on regulatory elements with multiple TF binding sites. For clarity, we limited the presentation of these data to one DNA strand only. Ectopically activated (or derepressed) TSSs after TF knockdown were preferentially upstream of the targeted TF's binding site but downstream of the second TF (Fig. 4g (black triangle) and Extended Data Fig. 9b–d), corroborating the role of TF-mediated blocking in TSS selection. As expected, downregulated TSSs were found at the preferred distance downstream of the targeted TF binding site but, importantly, also when the site was preferentially positioned upstream of the second activator TF binding sites, for example, Sp1 (Fig. 4g (red ellipse) and Extended Data Fig. 9b–d). This emphasizes that NRF1 or NFY binding upstream of Sp1 is necessary for proper initiation strength, supporting an additive model for TF contribution to transcription activity when multiple TFs are bound at their preferential positions[15]. Together with the result that NRF1 or NFY binding downstream of Sp1 can repress transcription initiation, this finding demonstrates that the order and spacing among TFs is critical for cooperative or antagonistic function and TSS selection (Fig. 4h).

## TF positioning in human disease

To investigate the role of position-dependent TF function across human individuals and putative functions in disease, we analysed nascent TSSs captured in lymphoblastoid cell lines (LCLs) from 67 Yoruba individuals[63] with sequenced genomes. We first defined TSS quantitative trait loci (tssQTLs) by correlating SNPs with individual TSS levels, and then stratified the impact of tssQTLs in TF binding sites in a distance-dependent manner. This analysis revealed the same grouping of TFs into pure transcriptional activators, pure transcriptional repressors and dual-function TFs with distinct positional preferences, relative to the TSS, as observed in the mouse strain data (Fig. 2f), and often

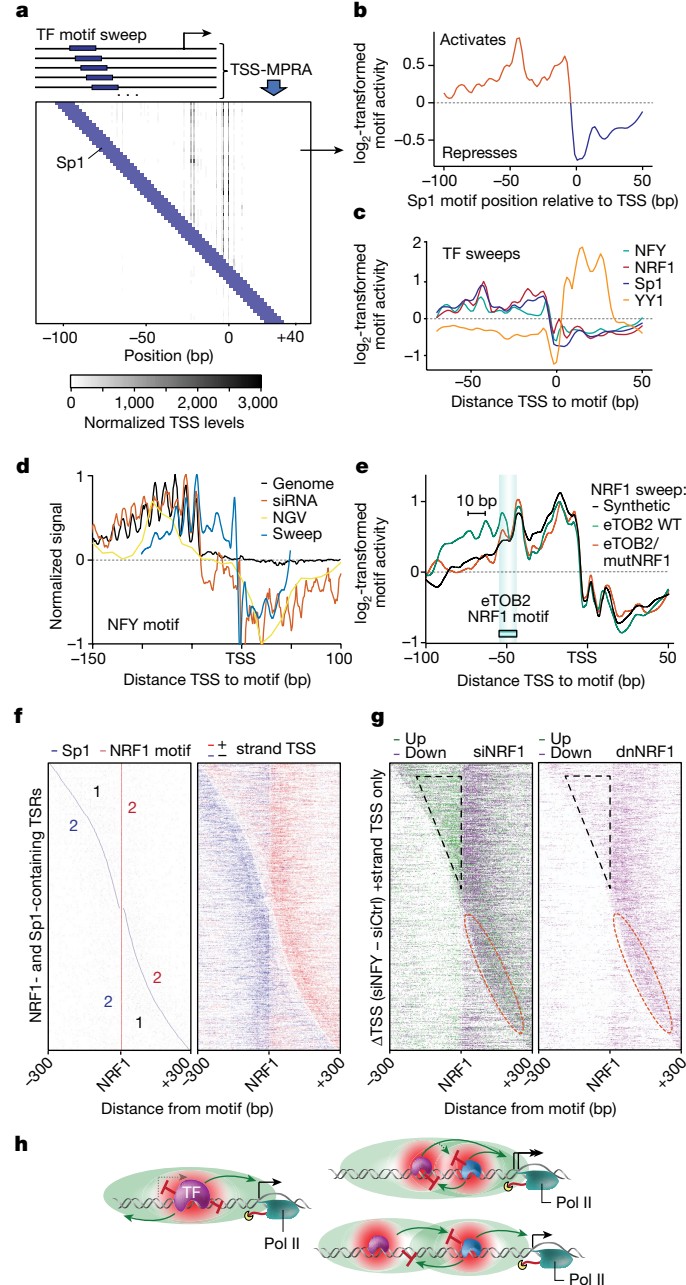

**Fig. 4 | Spatial interactions between TFs determines TSS position and initiation frequency. a**, Heat map of TSS measurements captured by TSS-MPRA for a 2 bp sweep of Sp1 binding sites (blue) from −100 to +40 across an artificial, motif-depleted promoter. **b**,**c**, Position-dependent activity patterns determined using TSS-MPRA TF sweeps resemble each TF's natural enrichment profile. The average $\log_2$-transformed change in initiation activity for all TSSs compared with their mean levels were plotted relative to the Sp1-binding-site (**b**) or the NFY-, NRF1- or YY1-binding-site (**c**) distance to each TSS. Average of $n = 2$ biological replicates. **d**, Multiple independent experimental approaches show similar spatial–functional profiles for a TF. Natural NFY-binding-site enrichment relative to the active TSS (black), the impact of *NFY* knockdown (orange), the impact of natural genetic variation in the NFY binding sites (green) and a TSS-MPRA NFY sweep (blue) reveal consistent, position-dependent functional profiles and superhelical preferences for NFY (each profile was minimum/maximum scaled to 1/−1). **e**, Position-dependent TF activity was altered when NRF1 is swept through the putative TOB2 enhancer (eTOB2) versus the TOB2 enhancer with the endogenous NRF1 binding site mutated (mutNRF1). **f**, Transcription initiation is affected by the relative spacing between TF binding sites and TSS location. TSRs containing both NRF1 and Sp1 binding sites sorted by the distance between them are shown with csRNA-seq initiation levels on both the positive (red; +) and negative (blue; −) strands. **g**, Position-dependent TF–TF interactions. TSSs upstream of the NRF1 but downstream of Sp1 (black triangle) are upregulated after *NRF1* knockdown while most downregulated TSSs are downstream from NRF1, even if Sp1 is found downstream as well (red circle). Upregulated TSSs are shown in green, and downregulated TSSs are shown in purple. Expression of dnNRF1 generally represses all nearby TSS activity. **h**, Model of how TF interactions can mediate TSS selection.

To corroborate these findings, we assessed the effect of mutating TF binding sites in 133 human promoters and enhancers on transcription initiation using TSS-MPRA. Consistently, mutation of TF binding sites within their naturally enriched positions was associated with repression, whereas mutations of sites outside of this region were associated with increased transcription initiation (Fig. 5d and Extended Data Fig. 10b,c). Moreover, mutation of these TF binding sites resulted in notable changes in TSS selection and therefore alternative 5′ UTRs (Fig. 5e and Extended Data Fig. 10c), a characteristic feature of numerous diseases[39,65,66]. Mutation of TF binding sites near TSSs or within their naturally enriched positions had the strongest effect relative to sites that occur elsewhere (Extended Data Fig. 10d). Together, these findings show that the position-dependent function of TFs can impact human gene regulation in health and disease.

## Discussion

Multiple independent lines of functional evidence consistently replicate spatial–functional profiles of a TF, including (1) TF perturbation; (2) natural genetic variation; (3) TF-binding-site insertion or deletion screens in natural TSRs; (4) synthetic DNA TF sweep TSS-MPRAs; and (5) analysis of human individuals and GWAS variants. Importantly, functional profiles align with the enriched or depleted positions for TF binding sites relative to TSSs genome-wide, indicating that the naturally observed positioning of TF binding sites can reveal important information about position-dependent TF function. The fact that these patterns also emerge for cell-type- or stimulus-specific TFs relative to regulated TSSs (Extended Data Fig. 2f and Extended Data Fig. 5g,h) indicates that TSS mapping followed by spatial analysis of TF binding sites could provide valuable insights into TF functions when studying a wide range of biological systems, disease states and even different species. Position-specific enrichment of TF binding sites in specific disease contexts can also be observed for inflammatory regulators during COVID-19 or in flavivirus infection[67,68]. We speculate that these profiles reflect the combined impact of multiple regulatory mechanisms for each TF. For example, activator TFs may function to remodel chromatin or directly recruit the transcription machinery to broader regions relative to the TF's binding site, but binding of TFs to DNA

mimicked each TF binding site's enriched localization in human TSSs (Fig. 5a and Supplementary Fig. 3). Integrating data from genome-wide association studies (GWAS) further demonstrated a position-dependent effect of disease-associated genetic variants on TSS levels, contingent on the position of TF binding sites. For example, the variant rs11122174-T, which is linked to defects in haematopoiesis[64], disrupts an NRF1 site in the *TTC13* promoter, resulting in decreased initiation downstream of the mutated site and an increase in initiation from upstream TSSs (Fig. 5b and Extended Data Fig. 10a). Indeed, tssQTL variants that weaken binding sites of many dual-function TFs (Sp proteins, ETS, NFY, NRF1, CRE, ZBTB33, AP-1 and PU.1) upstream of TSSs predominantly decreased transcription initiation ($P < 0.00022$, Fisher's exact test). Conversely, variants enhancing the consensus of weaker binding sites were associated with increased initiation ($P = 3.8 \times 10^{-6}$, Fig. 5c). By contrast, consistent with position-dependent TF function, TF-binding-site-disrupting variants downstream of the TSS increased transcription ($P = 0.0019$). Strengthening of a downstream site by comparison had a lesser impact, potentially because TF binding to moderate-affinity sites already causes steric hindrance ($P = 0.35$; Fig. 5c).

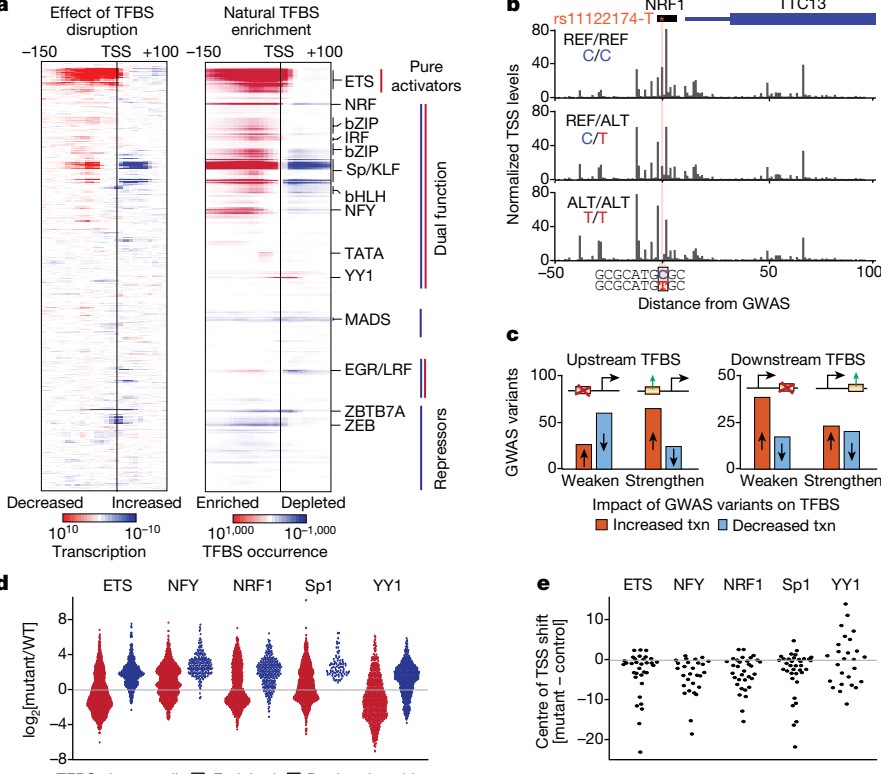

**Fig. 5 | Position-dependent TF function in human disease. a**, Positional TF binding site enrichment (right) and position-dependent activity of TFs based on the analysis of genetic variants and TSS activity (left) in LCLs across 67 human individuals calculated for 463 human TF motifs (30 bp windows evaluated at 10 bp increments). Statistical analysis was performed using two-sided Mann–Whitney *U*-tests or Fisher's exact tests with Benjamini–Hochberg correction. Note that only a subset of TFs with motifs is expressed in LCLs. A detailed map with all TFs annotated is provided in Supplementary Fig. 3. **b**, Disease-associated variants, identified through GWAS, recapitulate position-dependent TF function. Example of variant rs11122174, for which a C to T mutation disrupts a consensus NRF1 binding site leading to a general increase in upstream TSS activity and decrease in downstream TSS activity. **c**, Summary of the effect of disease-associated GWAS variants grouped by position relative to the TSSs suggests a role for position-dependent TF function in disease. GWAS variants

weakening TF binding sites upstream of TSSs were associated with reduced initiation while those strengthening TF binding sites were associated with increased transcription (txn). Vice versa, weakening TF binding sites increased proximal TSSs, consistent with the reported position-dependent TF blocking function. **d**, An analysis of 133 human promoters and enhancers using TSS-MPRA showed that the mutation of TF binding sites for ETS1, NFY, NRF1, Sp1 and YY1 within their naturally enriched position is associated with reduced initiation, while mutation of sites in positions at which the TF binding site is naturally depleted were associated with activation (≥20 promoters each). Each point represents an individual TSS. **e**, Mutation of TF binding sites resulted in changes in TSS selection and alternative 5′ UTRs. The mean shift of all TSS positions between the mutant and control elements is plotted for the mutation of each TF binding site family.

itself can also sterically hinder formation of or redirect the preinitiation complex in close proximity. Overall, these data indicate that TFs cooperatively drive TSS selection in a manner consistent with their unique position-dependent functional properties. As such, different spatial arrangements of the same sets of TFs can lead to distinct gene regulatory outcomes (Fig. 4h). More broadly, our findings highlight a spatial grammar as central to encode the multiple, often overlapping gene regulatory programs in our genome.

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

# Methods

## Experimental methods

**Cell culture, siRNA and mRNA transfections.** U2OS, HepG2, HEK293T or Vero E6 cells were grown at 37 °C with 5% $CO_2$ in DMEM (Cellgro) supplemented with 10% FBS (Gibco), 50 U penicillin and 50 µg streptomycin per ml (Gibco). For siRNA transfection, cells were washed once with PBS (Gibco), trypsinized for about 5 min and then washed twice with PBS (Gibco) by centrifugation at 400g for 5 min. Subsequently, about 3 million cells were resuspended in 150 µl gene pulser electroporation buffer (Bio-Rad) and siRNAs in 1× siRNA buffer (60 mM KCl, 6 mM HEPES pH 7.5, 0.2 mM $MgCl_2$). The following siRNAs were used: M-011796-02-0005 siGENOME human *YY1* (7528) siRNA, M-017924-01-0005 siGENOME human *NRF1* (4899) siRNA and siGFP (CCACUACCUGAGCAC CCAGU[69]) as a control. For transfection of the dominant-negative *NRF1* mutant (AA1-304)[45], the sequence was cloned into pGEM-T and mRNA was synthesized using the mMESSAGE mMACHINE T7 Transcription Kit (Thermo Fisher Scientific). siRNAs and dnNRF1 were used at a final concentration of 10 µM. The mixture was transferred to a 0.2 cm cuvette and pulsed once at 250 V for a constant 20 ms. After electroporation, 1 ml complete growth medium was added and cells grown for 24 h in a 6 cm dish. To collect RNA, the plates were washed three times using ice-cold PBS with 1 ml TRIzol reagent added. Cells were scraped and RNA was extracted as described by the manufacturer with addition of 1 µl 15 mg ml$^{-1}$ GlycoBlue coprecipitant (Thermo Fisher Scientific).

**Mouse BMDMs.** This study used total RNA originally collected from macrophages generated from C57BL/6 and SPRET strains of mice as part of a previous study. The derivation of BMDMs and their treatment with KLA for 1 h were performed as described previously[50]. Total RNA from these samples was used to perform csRNA-seq as described below.

**Western blot.** Cells were lysed in 1× NuPAGE LDS sample buffer (Thermo Fisher Scientific), sonicated for about 30 s and then incubated at 95 °C for 5 min under 2,000 rpm agitation. The samples were then centrifuged for 5 min at 21,000g and about 10 µg total protein was loaded on a NuPAGE 4 to 12% Bis-Tris gel (NP0321BOX, Thermo Fisher Scientific). The gel was washed for 5 min in water, transferred for 90 min at 200 mA in 1× NuPAGE transfer buffer (NP0006, Thermo Fisher Scientific) and blocked for 40 min in ~1% non-fat dry milk in TBS. Original data are shown in Supplementary Fig. 4. Primary antibodies were allowed to bind at 4 °C overnight. The following antibodies were used: anti-YY1 (Santa Cruz, sc-7341, HRP YY1 (H-10), 1:200), anti-NRF1 (Abcam, ab55744, 1:1,000) and anti-β-Actin (Cell Signaling, D6A8, 8457S, rabbit monoclonal antibody, 8457, 1:2,500). Western blots were quantified using Fiji (v.1.53j) with mean grey value only.

**csRNA-seq.** csRNA-seq was performed as described previously[21]. Small RNAs of around 20–60 nucleotides were size-selected from 0.4–2 µg of total RNA by denaturing gel electrophoresis. A 10% input sample was taken aside and the remainder enriched for 5′-capped RNAs. Monophosphorylated RNAs were selectively degraded by 1 h incubation with Terminator 5′-phosphate-dependent exonuclease (Lucigen). Subsequently, RNAs were 5′-dephosporylated through 90 min incubation in total with thermostable QuickCIP (NEB) in which the samples were briefly heated to 75 °C and quickly chilled on ice at the 60 min mark. Input (sRNA) and csRNA-seq libraries were prepared as described previously[70] using RppH (NEB) and the NEBNext Small RNA Library Prep kit, amplified for 14 cycles and sequenced single-end for 75 cycles on the Illumina NextSeq 500 system.

**ChIP–seq.** ChIP–seq was performed as described previously[71]. A total of $3 × 10^6$ U2OS cells was fixed for 10 min with 1% formaldehyde/PBS, the reaction was quenched by adding 2.625 M glycine, cells washed twice with ice-cold PBS, snap-frozen in 1 million cell pellets and stored at −80 °C. For dnNRF1–HA ChIP–seq, cells were collected 12 h after electroporation with 3 µg IVT *dnNRF1-HA* mRNA as described above. Fixed cells were thawed on ice, resuspended in 500 µl ice-cold buffer L2 (0.5% Empigen BB, 1% SDS, 50 mM Tris/HCl pH 7.5, 1 mM EDTA, 1 × protease inhibitor cocktail) and chromatin was sheared to an average DNA size of 300–500 bp by administering 7 pulses of 10 s duration at 13 W power output with 30 s pause on wet ice using a Misonix 3000 sonicator. The lysate was diluted 2.5-fold with 750 µl ice-cold L2 dilution buffer (20 mM Tris/HCl pH 7.5, 100 mM NaCl, 0.5% Triton X-100, 2 mM EDTA, 1× protease inhibitor cocktail), 1% of the lysate was kept as ChIP input, and the remainder was used for immunoprecipitation using the following antibodies: dnNRF1-HA (2 µg anti-HA, Abcam, ab9110), NRF1 (2 µg, Abcam, ab175932) and YY1 (2 µg, ActiveMotif, 61980) and 20 µl of Dynabeads Protein A while rotating overnight at 8 rpm and 4 °C. The next day, the beads were collected on a magnet, washed twice with 150 µl each of ice-cold wash buffer I (10 mM Tris/HCl pH 7.5, 150 mM NaCl, 1% Triton X-100, 0.1% SDS, 2 mM EDTA), wash buffer III (10 mM Tris/HCl pH 7.5, 250 mM LiCl, 1% IGEPAL CA-630, 0.7% deoxycholate, 1 mM EDTA) and TET (10 mM Tris/HCl pH 7.5, 1 mM EDTA, 0.2% Tween-20). Libraries were prepared directly on the antibody/chromatin-bound beads. After the last TET wash, the beads were suspended in 25 µl TT (10 mM Tris/HCl pH 7.5, 0.05% Tween-20), and libraries were prepared using NEBNext Ultra II reagents according to the manufacturer's protocol but with reagent volumes reduced by half, using 1 µl of 0.625 µM Bioo Nextflex DNA adapters per ligation reaction. DNA was eluted and cross-links were reversed by adding 4 µl 10% SDS, 4.5 µl 5 M NaCl, 3 µl EDTA, 1 µl proteinase K (20 mg ml$^{-1}$), 20 µl water, incubating for 1 h at 55 °C, then overnight at 65 °C. DNA was cleaned up by adding 2 µl SpeedBeads 3 EDAC in 62 µl of 20% PEG 8000/1.5 M NaCl, mixing and incubating for 10 min at room temperature. SpeedBeads were collected on a magnet, washed twice by adding 150 µl 80% ethanol for 30 s each, collecting beads and aspirating the supernatant. After air-drying the SpeedBeads, DNA was eluted in 25 µl TT and the DNA contained in the eluate was amplified for 12 cycles in 50 µl PCR reactions using the NEBNext High-Fidelity 2× PCR Master Mix or the NEBNext Ultra II PCR Master Mix and 0.5 µM each of primers Solexa 1GA and Solexa 1GB. Libraries were cleaned up as described above by adding 36.5 µl 20% PEG 8000/2.5 M NaCl and 2 µl Speedbeads, two washes with 150 µl 80% ethanol for 30 s each, air-drying beads and eluting the DNA into 20 µl TT. ChIP library size distributions were estimated after 2% agarose/ TBE gel electrophoresis of 2 µl of library, and library DNA amounts measured using the Qubit HS dsDNA kit on the Qubit fluorometer. ChIP input material (1% of sheared DNA) was treated with RNase for 15 min at 37 °C in EB buffer (10 mM Tris pH 8, 0.5% SDS, 5 mM EDTA, 280 mM NaCl), then digested with proteinase K for 1 h at 55 °C and cross-links were reversed at 65 °C for 30 min to overnight. DNA was cleaned up using 2 µl SpeedBeads 3 EDAC in 61 µl of 20% PEG 8000/1.5 M NaCl and washed with 80% ethanol as described above. DNA was eluted from the magnetic beads with 20 µl of TT and library preparation and amplification were performed as described for the ChIP samples. Libraries were sequenced single-end for 75 cycles to a depth of 20–25 million reads on the Illumina NextSeq 500 instrument.

**TSS-MPRA.** *Twist Bioscience insert library cloning.* Insert library (10 ng) was PCR amplified in 50 µl mastermix (25 µl Q5 2× MM, 0.25 µl 100 µM pMPRA1-LH (5′-GGTAACCGGTCCAGCTCA), 0.25 µl 100 µM pMPRA1-RH (5′-CGTGTGCTCTTCCGATCT)) under the following conditions: 30 s at 98 °C; followed by 2× cycles of 10 s at 98 °C, 20 s at 65 °C and 15 s at 72 °C; followed by one more cycle with access primers and finally 2 min at 72 °C. The PCR product (~200 bp) was run on a 2% agarose gel, then gel-extracted and cleaned up using PureLink Quick Gel Extraction Kit (Invitrogen; K210012). The library pool was Gibson-assembled into BsaI-linearized pTSS-MPRA-Empty plasmid[57]. This reporter plasmid is based on the background-reduced pGL4.10 reporter backbone (Promega), with a cloning site harbouring the 18 bp pMPRA1-LH sequence

followed by tandem BsaI sites for linearization and the TruSeq Read 2-compatible pMPRA1-RH sequence and a downstream landing site for reverse transcription primer RS2, and an eGFP ORF replacing the luciferase 2 gene. pMPRA1 (400 ng) was digested with BsaI in 20 µl (1 µl BsaI, 2 µl CutSmart buffer (NEB)) at 37 °C for 1 h and linearized plasmid was gel-extracted from an agarose/TBE gel using the PureLink Quick Gel Extraction Kit (Invitrogen; K210012). Amplified library was Gibson-assembled into cut plasmid using NEBuilder HiFi 2× master mix with a fivefold molar excess of the library at 50 °C for 1 h in a 4 µl total volume.

*Transformation.* NEB 5-alpha (10 µl) chemically competent *Escherichia coli* (high efficiency, NEB, C2988) were transformed with 1 µl of Gibson assembly reaction, mixed and placed on ice for 30 min. Cells were subjected to heat-shock at 42 °C for 30 s, and then placed onto ice again for 5 min. A total of 950 µl of room temperature SOC was added and the cells were then incubated at 37 °C for 1 h while mixing at 250 rpm. Then, 200 µl of SOC culture was added to 200 ml of LB broth containing 100 µg ml⁻¹ carbenicillin and agitated at 37 °C, 225 rpm for 16–18 h before isolating plasmids using the PureLink HiPure Plasmid Maxiprep Kit (Invitrogen; K210006). Note that, for higher-diversity libraries such as fragmented genomes or scrambled oligos, precipitate your assembled plasmid and use electroporation.

*Transfection.* About 800,000 HEK293T cells were seeded into 3 ml DMEM (Cellgro, supplemented with 10% FBS (Gibco), 50 U penicillin and 50 µg streptomycin per ml (Gibco)) 24 h before transfection in a 6 cm dish and grown at 37 °C with 5% $CO_2$. For each plate and construct, 25 µg plasmid DNA was transfected using Lipofectamine 3000 (Thermo Fisher Scientific) as described by the manufacturer. After 8 h, cells were washed three times with PBS (Gibco) and RNA was extracted using 1 ml Trizol (Thermo Fisher Scientific).

*5′-capped RNA enrichment.* In total, 10–15 µg RNA (one-third of the total) was dissolved in 15 µl TE′T (10 mM Tris pH 8.0, 0.1 mM EDTA, 0.05% Tween-20), heated to 75 °C for 90 s, then quickly chilled on wet ice. Non-capped RNAs were dephosphorylated with Quick CIP (NEB) and DNA digested by adding 25.25 µl MM1 (double-distilled $H_2O$ + 0.05% Tween-20, 5 µl CutSmart buffer, 0.75 µl SUPERase in RNase inhibitor (20 U µl⁻¹, Thermo Fisher Scientific), 0.5 µl RQ1 DNase (Promega), 2 µl Quick CIP). The sample was mixed well and incubated at 37 °C for 90 min. For more complete dephosphorylation of uncapped transcripts, RNA was denatured a second time in MM1 at 75 °C for 30 s in a prewarmed water bath and then quickly chilled on ice for 2 min before incubating the sample at 37 °C for another 30 min. After double cap-enrichment, RNA was purified by adding 500 µl Trizol LS, mixed and then adding 140 µl TE′T and 140 µl $CHCl_3$ + IAA (24:1, Sigma Aldrich). The samples were vortexed vigorously and subsequently centrifuged for 10 min at 12,000g at room temperature (21 °C; allowing the CIP to move to the lower phase). After phase separation, the top layer was taken, and RNA was precipitated by mixing first with 1/10 vol 3 M NaOAc and then 1 vol 100% isopropanol. The mixture was incubated at −20 °C for at least 20 min to overnight, then RNA precipitated by centrifuging at >21,000g for 30 min at 4 °C. The supernatant was removed, the samples centrifuged once more and all of the remaining liquid was removed before washing the pellet in 400 µl 75% ethanol by inversions followed by centrifugation as before. All ethanol was completely removed, and the pellet was air-dried at room temperature for around 3–5 min.

*Library preparation.* RNA pellets were resuspended in 5 µl TE′T, heated 75 °C for 90 s, then quickly chilled on ice. To remove 5′ caps, we next added 10 µl DecappingMM (3.25 µl double-distilled $H_2O$ + 0.05% Tween-20, 1.5 µl 10× T4 RNA ligase buffer, 4 µl PEG 8000, 0.25 µl SUPERase in RNase inhibitor and 1 µl RppH (NEB)) and incubated the samples at 37 °C for 1 h. After decapping, 5′ adapters were ligated by T4 RNA ligase 1 using 10 µl L1MM (1 µl 10× T4 RNA ligase buffer, 2 µl 10 mM ATP, 1.5 µl 10 µM 5′ adapter, 4.5 µl PEG 8000 and 1 µl T4 RNA ligase 1 (NEB)) and incubating at room temperature (21 °C) for 2 h. For better results, we

next repeated the Trizol clean-up as described in the '5′-capped RNA enrichment' section.

RNA pellets were next resuspended in 7 µl annealing MM (1 µl 20 mM RS2 primer (5′-AGCGGATAACAATTTCACACAGGA-3′), 2 µl 700 mM KCl and 4 µl TET (10 mM Tris pH 7.5, 1 mM EDTA, 0.05% Tween-20)) and incubated at 75 °C for 90 s followed by 30 min at 56 °C and then cooled down to room temperature. Next, 13 µl reverse transcription MM (7.5 µl double-distilled $H_2O$ + 0.05% Tween-20, 1 µl reverse transcriptase buffer (homebrew, 500 mM Tris-HCl pH 8.3, 30 mM $MgCl_2$), 2 µl 0.1 M DTT, 2 µl 10 mM dNTPs, 0.5 µl SUPERase in RNase inhibitor and 1 µl Protoscript II (NEB)) were added and samples incubate at 50 °C for 1 h.

The samples were amplified (95 °C for 3 min; then 14 cycles of 95 °C for 30 s, 62 °C for 30 s, 72 °C for 45 s; and 72 °C for 3 min) at 50 µl volume (25 µl 2× Q5 MM (NEB), 2.8 µl 5 M betaine (Sigma-Aldrich), 2 µl 10 µM 3′ barcoding primer, 0.2 µl 100 µM 5′ primer). Note that the extended extension time is important to ensure that all PCR products are completely amplified. After PCR, 1 µl of 20 mg ml⁻¹ RNase A was added and the reaction was incubated at 37 °C for 30 min. PCR reactions were purified using 1.5 vol of beads (2 µl Sera-Mag carbonylated beads (Cytiva), 34 µl 5 M NaCl, 37.5 µl 40% PEG 8000) and sequenced paired-end 50/30 on the NextSeq 500 system.

## Data analysis

**HOMER2.** HOMER2 enables investigation of DNA sequence fragments and TF binding sites enrichment accounting for the general nucleotide content of the input fragments (for example, total GC content) and position-dependent nucleotide biases (Extended Data Fig. 1 and Supplementary Fig. 5). While HOMER2 is used to analyse sequences and account for sequence biases near TSSs, similar to its predecessor HOMER[23], the software package can be used for a wide range of data analysis. HOMER2 is available online (http://homer.ucsd.edu/homer2/; HOMER2 is fully integrated into HOMER starting with v.5).

*Position-dependent background selection and motif enrichment in HOMER2.* To account for the influence of position-dependent sequence bias on the calculation of TF binding motif enrichment, we developed improvements in HOMER to select random genomic sequences that contain the same position-dependent sequence features present in a set of target sequences (for example, sequence anchored at thee TSS). Previously, when HOMER performed motif-enrichment calculations, random sequences from the genome were selected to construct an empirical background set such that the overall distribution of GC% per sequence in the background set matched that of the target sequence set (Extended Data Fig. 1). The primary purpose was to address sequence biases present in regulatory elements that overlap CpG islands, which have different overall sequence characteristics from other regions of the genome. HOMER2 can now select background sequences that preserve positional preferences for nucleotide composition, while still matching the overall GC% fragment distribution in the dataset (Extended Data Fig. 1). Position-dependent nucleotide composition can be considered for different $k$-mer lengths, that is, $k = 1$ (simple nucleotide frequency), $k = 2$ (dinucleotide frequency), $k = 3$ (trinucleotide frequency) and so on. $k = 2$ was used in this study. This selection is restricted to datasets with a fixed sequence length to unambiguously assess position-specific information relative to a defined anchor point. In addition to the sequence-selection procedure outlined below, HOMER has also been updated to generate synthetic sequences based on a Markov model describing $k$-mer transitions in a position-dependent manner to create a background dataset of synthetic sequences with the desired characteristics.

First, target sequences (for example, ±200 bp from TSS locations) are assessed for their overall GC% content and positional $k$-mer content (Supplementary Fig. 5). Target sequences are then sorted by their overall GC% and segregated into $n$ bins corresponding to increasing ranges of overall GC% content ($n = 10$ for this study). Sequences from the genome matching these GC% ranges are identified and putatively

assigned to the appropriate GC-bin. Next, the genomic sequences assigned from each bin are then sampled to generate a set of background sequences that matches the positional $k$-mer frequencies of the target sequences in each corresponding GC bin. Background sequence selection in each GC-bin is performed using an iterative gradient descent approach that progressively removes sequences until the final desired number of background sequences per bin is reached. During each iteration, the overall similarity between the positional $k$-mer frequency in the target and background sets is computed. Each target and background sequence is then scored against these $k$-mer frequency differences based on the $k$-mers present in each sequence at each position and adding the differences in their frequencies (linear combination). Background sequences for the next iteration are then randomly sampled on the basis of the relative fraction of target sequences that have a similar overall difference score, which attempts to match the same overall distribution of the target sequences. This process is repeated until the differences in $k$-mer frequencies approaches zero or a maximum number of iterations is reached. Once the iterative selection process for background sequences in each GC bin is complete, the sequences are combined into a final background sequence set and the distribution of the overall GC% and position-dependent $k$-mer frequencies are calculated and compared to the original target sequences (Supplementary Fig. 5). These sequences can then be used to more accurately consider TF motif enrichment (below) or be exported for other applications.

To calculate motif enrichment, target and background sequences are scanned for motif matches using HOMER to generate a complete table of motif occurrences. For any given interval (for example, −50 to −40 relative to the TSS), the total number of target and background motif occurrences within that range are tabulated and their enrichment is calculated using the Fisher exact test (hypergeometric distribution). In cases in which there are proportionally less motif occurrences in the target sequences compared with the background, the depletion is noted and $1 − P$ is reported. This is performed over all positional ranges and for each motif queried, and the resulting $P$ values are further corrected for multiple-hypothesis testing using the Benjamini−Hochberg method. The corrected log-transformed $P$ values are then reported, with comparisons containing depleted values reported as $−1 × \log[P]$ to reflect depletion (versus $1 × \log[P]$ for enrichment). De novo motif enrichment is performed by applying the original HOMER search algorithm using background sequences generated by HOMER2.

*Motif analysis.* To unbiasedly identify the most strongly enriched or depleted TF binding sites associated with transcription initiation (Fig. 1 and Extended Data Figs. 2 and 3), TSRs from U2OS cells were analysed from −150 to +100 bp relative to the TSS using HOMER2 positionally matched sequences from random genomic regions (GCbins = 10, kmer = 2). Maps of known motif enrichment were calculated for all 463 TF motifs in the HOMER known motif database for each strand separately and at each bp using 3 bp windows (Fig. 1d). Enrichment at human LCL TSSs was calculated on both strands every 10 bp using 30 bp windows (Fig. 5a) to directly compare with the analysis based on genetic variants (Fig. 5a). Motif heat maps were created by clustering the log-transformed $P_{adj}$ values using the correlation coefficient as a distance metric (Cluster v.3.0)[72] and visualizing the resulting heat map using Java TreeView[73] (v.1.1.6r4). Motif occurrences, including histograms showing the density of binding sites relative to the TSS (reported as motifs per bp per TSS), and the average nucleotide frequency was calculated using HOMER's annotatePeaks.pl tool and visualized with Java TreeView, Excel (v.16.83) or R (v.4.2.2). TSSs were assigned as a promoter TSS if their position was on the same strand and within 200 bp of a GENCODE annotated TSS. Promoter antisense TSSs were defined as those on the opposite strand in the range of −400 to +200 relative to a GENCODE-annotated TSS. Promoter-distal (for example, enhancer) TSSs were defined as those that are found greater than 1 kb from a GENCODE-annotated TSS. Spectral analysis of TF binding sites

was performed on TF binding sites found between −120 and −40 bp relative to the TSS, corresponding to the region where many TFs appear to exhibit cyclical patterns of positional preference. The power spectrum (Fourier analysis) was calculated for periods from 0 to 50 bp in 0.1 bp increments on each TF's strand-specific binding profile. The resulting power spectra were normalized to their maximum value to facilitate comparison. To segregate TSSs on the basis of the presence of initiator core promoter elements, the genomic sequence adjacent to each TSS was scanned for a strand and position-specific match to the sequence IUPAC motif BBCA + 1BW (where the A + 1 defines the initiating nucleotide)[74]. TSSs with a match were considered to be Inr-containing TSSs.

*Analysis of TSSs and TF binding sites in the context of natural genetic variation.* To assess how variation in TF binding site sequences relate to changes in TSS activity, we developed a framework for natural genetic variation analysis within HOMER2 that was inspired by MAGGIE[51]. First, variants in *cis* near TSSs with significant changes in activity are found (for example, <200 bp) and the alleles associated with higher activity are assigned as 'active', while their corresponding alternative alleles are assigned as 'inactive'. If a variant overlaps a given TF binding site, the change in motif log odds score is calculated for that motif (active − inactive) and a distribution of motif score changes is created for all variants impacting that motif at sites within a given distance interval from the TSS. In MAGGIE's original formulation, the null assumption underlying the nonparametric rank-sum significance calculations was that changes in motif score are independent of TSS activity, implying that the average of the distribution of motif scores should be zero. However, variants found near TSSs with differential activity do not follow a uniform pattern and may impact other sequence features that influence transcription initiation in a position-dependent manner (for example, core promoter elements) (Fig. 1a and Extended Data Fig. 5e,f). This implies that the expected changes in log odds scores for a given motif at a given distance from the TSS may follow a different distribution (that is, !=0).

To more accurately assess how genetic variation impacts TF binding sites and TSS activity as a function of distance to the TSS, HOMER2 attempts to model the expected distribution of changes in motif log odds scores given the full distribution of variants observed relative to the TSS. This analysis is limited to single-nucleotide variants (SNVs/SNPs), which are evaluated independently from one another. First, a saturation mutagenesis scan is performed on each sequence to identify all of the positions where a match to a given motif may occur, and all of the potential differences in log odds scores as function of the variant and position are recorded. Then, for a given interval, an expected distribution of motif score changes is constructed taking the changes observed in the saturation mutagenesis analysis and then scaling their expected frequency by the total number of variants of each type (that is, A to C) observed at that position relative to the TSS. This expected distribution is then compared to the observed changes in motif scores from the actual variants and their sequences using nonparametric rank-sum tests (Mann−Whitney $U$-tests) to calculate the significance of the difference. This analysis is analogous to randomizing the sequences containing each variant while preserving the position and nucleotide identity of the variant relative to the TSS. After all motifs are evaluated at all intervals, the resulting $P$ values are then corrected for multiple-hypothesis testing using the Benjamini−Hochberg method. Average changes in motif score at each position and overlap with GWAS-annotated variants are also reported.

For this study, the analysis was performed for all 463 motifs in the HOMER2 known-TF-binding-site library using 30 bp overlapping windows evaluated every 10 bp from −150 to +100 bp relative to the TSS. Larger windows improve the sensitivity by capturing more binding sites of a given TF motif with DNA variants between strains to increase the sensitivity, while smaller regions improve the resolution of the analysis. When analysing differences between strains of mice, active alleles were defined by TSSs that were significantly differentially regulated

between strains (see below). For analysing human variants, active alleles were defined on the basis of a positive slope from significant tssQTLs ($|slope| > 0.1$, $P < 0.25$).

As an alternative approach, we performed a second analysis by directly comparing the sequences found in each mouse strain's genome assembly using the original MAGGIE program (v.1.1.1, https://github.com/zeyang-shen/maggie; Supplementary Fig. 2c). This approach differs from that above in that it can assess structural variation and indels in addition to SNVs, but does not model the position-dependent changes in expected motif score differences due to the arbitrary types of sequence variation considered. For this alternative analysis, sequences from −150 to +100 bp relative to the differentially regulated TSS (>1.5 shrunken fold change in one mouse strain versus the other) were analysed. We applied MAGGIE multiple times using an overlapping windowed approach to analyse genomic sequences associated with specific distance intervals from the TSS, similar to our approach above. To perform the MAGGIE calculation for each windowed region and TF binding site, sequences corresponding to a given region relative to the TSS were extracted from either the C57BL/6/mm10 genome or from regions in the same distance range relative to the homologous TSS position in the SPRET genome. These regions were then scanned using HOMER to identify TSSs associated with a match to the given motif in at least one of the two mouse strains. These sequences were then analysed using MAGGIE to identify pairs for which strain specific mutations were associated with changes in TSS activity. The significance was reported as the $\log[P]$ reported by Maggie, which was then signed to indicate whether the association was more strongly associated with increasing (negative $\log[P]$ values) or decreasing (positive $\log[P]$ values) TSS activity. TF-binding-site enrichment heat maps were generated by combining signed $\log[P]$ value results across all TF binding site and TSS-motif distances and then clustering the values using the correlation coefficient (Cluster 3.0) and visualizing the resulting heat map using Java TreeView.

*TF–TF spacing and transcription initiation analysis.* To analyse patterns of transcription initiation near pairs of TF binding sites (Fig. 4f), non-redundant binding sites for the first TF binding sites were first identified by scanning all TSRs from −300 bp to +300 bp using HOMER2. These sites were then scanned a second time from −300 bp to +300 to identify instances of the second TF binding sites, and each region containing both TF binding sites was then sorted based on the position of the second TF binding sites relative to the first. Note that, if multiple instances of one of the binding sites are found in the vicinity of the other sites, these regions will be represented multiple times in the list. The sorted regions, centred on the first TF binding site, were then used to generate TF initiation level heat maps using HOMER's annotatePeaks.pl program using the parameter '-ghist'.

*MEIRLOP score-based motif analysis.* To analyse how well each TF binding site associates with the activity level of TSS, we used MEIRLOP (v.0.0.16)[75], analysing motif occurrences from −150 to +50 bp relative to the TSS and associating them with the total count of csRNA-seq reads ($\log_2$). MEIRLOP assesses the dinucleotide content of each sequence and models them as covariates when performing logistic regression. Statistically significant enrichment coefficients ($P_{adj} < 0.05$) were reported along with their confidence intervals (Extended Data Fig. 2b; https://github.com/npdeloss/meirlop).

*MEPP positional enrichment scoring.* To analyse how the spatial distribution of a TF binding site associates with changes in TSS activity, we used Motif Enrichment Positional Profiles (MEPP, v.0.0.1)[76]. For a given motif PWM (for example, NRF1 motif), MEPP assesses the positional enrichment of the motif relative to a set of scored sequences (for example, the $\log_2$-transformed fold change between the control siRNA and *NRF1* siRNA conditions) that are anchored by a key feature (for example, the TSS). MEPP first calculates the positions of the TF binding site across all regions, generating a heat map to visualize the locations and PWM log-transformed odds scores (Fig. 1e (middle)). Positions are

indicated relative to the centre of the PWM motif. At each motif position surrounding the key anchor feature, we calculate the partial Pearson correlation of a sequence's score with the motif's PWM log-odds score at that position (while controlling for GC content as a covariate in the calculation). The positional correlation is then presented as a profile below the heat map (Fig. 1e (bottom); https://github.com/npdeloss/mepp).

**csRNA-seq analysis.** Genome originating TSS location and activity levels were determined using csRNA-seq and generally analysed using HOMER[21,23]. Additional information, including analysis tutorials are available online (http://homer.ucsd.edu/homer/ngs/csRNAseq/index.html).

csRNA-seq (small capped RNAs, ~20–60 nucleotides) and total small RNA-seq input sequencing reads were trimmed of their adapter sequences using HOMER ('homerTools trim -3 AGATCGGAAGAG CACACGTCT -mis 2 -minMatchLength 4 -min 20') and aligned to the appropriate genome (GRCh38/hg38, GRCm38/mm10) using STAR (v.2.7.10a)[77] with the default parameters. Only reads with a single, unique alignment (MAPQ ≥ 10) were considered in the downstream analysis. Furthermore, reads with spliced or soft clipped alignments were discarded to ensure accurate TSSs. The same analysis strategy was also used to reanalyse previously published Start-seq and PRO-cap TSS profiling data to ensure the data were processed in a uniform and consistent manner, although different adapter sequences were trimmed according to each published protocol.

Two separate transcription initiation analysis strategies were used in this study. In most cases, individual, single-nucleotide TSS positions were independently analysed. For a subset of analyses, we analysed transcription initiation levels in the context of TSRs, which comprise several closely spaced individual TSS. Individual TSS locations are useful for characterizing spacing relationships at single-bp resolution, whereas TSRs are more useful for describing the overall transcription activity at whole regulatory elements.

To analyse TSSs at the single-nucleotide resolution, the aligned position of the 5′ nucleotide of each csRNA-seq read was used to create a map of putative TSS locations in the genome. To ensure that we use high-quality TSSs that could be reliably quantified across different conditions, only TSS locations with at least 7 reads per $10^7$ total aligned reads across all compared replicates and conditions (for example, control siRNA, *NRF1* siRNA and so on) were retained for further analysis[21]. Furthermore, any TSS that had higher normalized read density in the small RNA input sequencing was discarded as a likely false-positive TSS location. These sites often include miRNAs and other high-abundance RNA species that are not entirely depleted in the csRNA-seq cap enrichment protocol. To quantify the change in TSS levels between conditions, a unified map of confident TSS positions is first determined across the set of experiments to be compared. Then, the TSS activity levels are assessed for each replicate and each experimental condition by first counting the raw read coverage across each TSS and all experiments and normalizing the dataset using DESeq2's rlog variance stabilizing transform (v.1.38.3)[78]. Changes in transcriptional activity were then reported as the $\log_2$-transformed fold change representing the difference between averaged rlog transformed activity levels across conditions (similar to a shrunken fold change). DESeq2 was used to identify significantly differentially regulated TSS, defined as TSS exhibiting a change of at least 1.5 fold and $P_{adj} < 0.05$, unless otherwise noted.

TSRs, representing loci with significant transcription initiation activity from one or more individual TSSs on the same strand from the same regulatory element (that is, peaks in csRNA-seq), were defined using HOMER's findcsRNATSR.pl tool, which uses short input RNA-seq, traditional total RNA-seq and annotated gene locations to find regions of highly active TSSs and then eliminate loci with csRNA-seq signals arising from non-initiating, high-abundance RNAs that nonetheless are captured and sequenced by the method (further details are available

in a previous study[21]). Replicate experiments were first pooled to form meta-experiments for each condition before identifying TSRs. Annotation information, including gene assignments, promoter distal, stable transcript and bidirectional annotations are provided by findcsRNATSS.pl. To identify differentially regulated TSRs, TSRs identified in each condition were first pooled (union) to identify a combined set of TSRs represented in the dataset using HOMER's mergePeaks tool using the option '-strand'. The resulting combined TSRs were then quantified across all individual replicate samples by counting the 5' ends of reads aligned at each TSR on the correct strand. The raw read count table was then analysed using DESeq2 to calculate normalized rlog-transformed activity levels and identify differentially regulated TSRs, similar to the analysis of TSSs[78].

In all cases, normalized genome browser visualization tracks for csRNA-seq data were generated using HOMER's makeUCSCfile program using the '-tss' option and visualized using either the UCSC Genome Browser[79] or IGV[80]. Annotation of TSS/TSR locations to the nearest gene was performed using HOMER's annotatePeaks.pl program using GENCODE as the reference annotation.

Additional information about csRNA-seq analysis and tips for analysing TSS data is available at the HOMER website (http://homer.ucsd.edu/homer/ngs/csRNAseq/index.html).

**Analysis of TSSs across two strains of mice.** To analyse how changes in genomic sequence impact TSS activity from two different strains of mice, we first took steps to ensure that TSS positions were conserved and detectable in both mouse strains to avoid analysing TSSs that may exhibit differential activity due to technical/analytical reasons, or TSSs found in non-homologous DNA. This was accomplished by ensuring that all csRNA-seq reads used in the analysis could be aligned to a single, unique location in the genomes of both mouse strains. Furthermore, the location that each csRNA-seq read aligns to in each genome must correspond to a homologous position in the full genome alignment, indicating that they represent the equivalent TSS positions in each strain. This latter filter helps to eliminate TSSs mapping to positions in repetitive or duplicated regions that may not be resolved in one or both genome assemblies.

To identify valid TSS positions for the natural genetic variation analysis, csRNA-seq and small input RNA-seq reads were first trimmed to remove adapter sequences. Reads from each mouse experiment (regardless of the strain of origin) were aligned to both the C57BL/6 (GRCm38/mm10) and SPRET (GCA_001624865.1/SPRET_EiJ_v1) genomes using STAR with the default parameters. Only reads that aligned to a single, unique location (MAPQ ≥ 10) were considered further. Next, TSS positions representing the 5' end of the reads were mapped to the other mouse strain's genome using UCSC's liftOver tool and the corresponding C57BL/6/SPRET liftOver files (http://hgdownload.cse.ucsc.edu/goldenpath/mm10/liftOver/mm10ToGCA_001624865.1_SPRET_EiJ_v1.over.chain.gz, https://hgdownload.soe.ucsc.edu/goldenPath/GCA_001624865.1_SPRET_EiJ_v1/liftOver/GCA_001624865.1_SPRET_EiJ_v1ToMm10.over.chain.gz). If the liftOver calculation yielded the same TSS location as the alignment from the other mouse strain, the read was retained for downstream analysis. Confident TSS locations, including DESeq2 rlog-normalized values and $\log_2$-transformed fold changes were then calculated based on the alignment positions reported in the mm10 genome as described in the sections above. Strain-specific differentially regulated TSSs used in the analysis of natural genetic variation were determined by DESeq2 using $P_{adj}$ of 0.25, resulting in 431,310 variant–TSS pairs for analysis.

**Analysis of tssQTLs from LCL PRO-Cap data.** *Variant file merging and filtering.* Per-chromosome VCF files containing genotyping data for the samples analysed previously[63] were downloaded from the 1000 Genome project (ftp://ftp.1000genomes.ebi.ac.uk/vol1/ftp/data_collections/1000G_2504_high_coverage/working/20201028_

3202_raw_GT_with_annot/). Using bcftools[81], these VCFs were then filtered for samples and variants observed from 67 individuals corresponding to PRO-cap samples from Gene Expression Omnibus (GEO) accession GSE110638. The variants in these VCFs were then normalized and named using their location and allelic data using bcftools. The per-chromosome VCFs were then merged, while trimming unobserved alternate alleles, removing sites without called genotypes and requiring minor allele counts of at least 1. These were further filtered to include only variants that were flagged as passing all upstream quality checks and were called with a minimum depth of 10. The resulting VCF was converted into PLINK format using plink2 (v.2.00a2.3LM)[82], retaining only SNPs with less than 50% of genotype calls missing, and a minor allele frequency greater than 0.05. A set of genotype principal component analysis (PCA) covariates was also generated using plink2.

*LCL PRO-Cap data processing.* To mitigate allele-specific alignment effects, a masked genome was created that set bases at the filtered SNP coordinates to Ns. We then trimmed adapter sequences from reads using fastp and aligned them to the masked genome using STAR using the default parameters. The resulting alignments were then processed using HOMER. Tag directories were then separated on the basis of whether they belonged to PRO-cap or PRO-seq experiments. To call TSSs, PRO-cap and PRO-seq reads were processed in the same manner as csRNA-seq and total small RNA-seq, respectively, to identify a unified set of human TSSs. We then obtained rlog-normalized from raw counts quantifying coverage of the 5' ends of reads from each sample, using HOMER annotatePeaks.pl. To avoid bias from extreme outliers, we retained only TSSs with a minimum count of 10 reads in 10 samples. To control for broad genome-wide expression variance, we obtained a set of 50 expression PCA covariates.

*tssQTL calling.* To call QTLs from TSS data, we used TensorQTL (v.1.0.3)[83] to determine the link between filtered SNP genotypes and TSS expression phenotypes, while controlling for covariance from sex, genotype PCA and TSS expression PCA. TensorQTL was run in both 'cis' and 'cis_nominal' modes, with a cis window of 300 bp. We then analysed each cis-nominal QTL SNP within the framework of our natural genetic variation analysis, limiting our analysis to tssQTLs with $P < 0.25$ and $|slope| > 0.1$, leading to a total of 194,746 variant–TSS combinations used in the analysis.

**ChIP–seq analysis.** ChIP–seq reads were aligned to the appropriate human genome (GRCh38/hg38) using STAR with the default parameters[77]. Only reads with a single, unique alignment (MAPQ ≥ 10) were considered in the downstream analysis. ChIP–seq peaks were determined using HOMER's findPeaks program using '-style factor'. Quantification of ChIP–seq reads associated with peaks, annotation to the nearest annotated gene TSS, calculation of TF binding site presence (−100 to +100 relative to the peak centre) and visualization of normalized read pileups for the genome browser were all conducted using HOMER. The same analysis strategy used for our dnNRF1, NRF1 and YY1 ChIP–seq data was also used to reanalyse TF ChIP–seq data from ENCODE[84] to ensure that the data were processed in a uniform and consistent manner.

**TSS-MPRA analysis.** Three different DNA insert designs were used for TSS-MPRA in this study. For each design, 400–500 DNA inserts were designed consisting of 155 bp of query DNA (described below) and redundantly coupled with 4 or 5 independent 11 bp barcodes optimized for their molecular properties and diversity[85] to generate a total of 2,000 DNA inserts per design. Genome-encoded TSR sequences queried by TSS-MPRA were designed to capture the sequence from −113 to +42 bp relative to the primary TSS to capture most of the upstream TF binding sites and core promoter region. Key sequences used in the MPRA design are reported in Supplementary Table 2, and full TSS-MPRA design files and sequences are available at the GEO (GSE199431).

To process TSS-MPRA results, raw RNA and DNA sequencing reads, corresponding to the RNA transcripts and input DNA library, respectively, were trimmed for the 5′ adapter sequence GGTAACCGGTC CAGCTCA on the R1 read using cutadapt v.3.4. The trimmed reads were aligned to the reporter library using STAR (v.2.7.10a)[77], specifying an option to preclude soft-clipping at the 5′ end of R1 (STAR --outSAMattributes All --genomeDir library/tfsweep.STARIndex --runThreadN 12 --readFilesIn file.fastq --outFileNamePrefix star/ Sweep_1. --alignEndsType Extend5pOfRead1 --outSAMtype BAM Sorted-ByCoordinate). For DNA plasmid control samples, the uniquely aligned read pairs were counted to later scale transcriptional output. For RNA samples, uniquely aligned read pairs were further processed to identify exact TSSs, yielding a matrix of start sites per sequence position. Any alignments showing mismatches at their 5′ ends were not counted and reporter sequences with fewer than 50 total DNA alignments were also ignored. Specific DNA insert details and analysis approaches tailored for each TSS-MPRA design are described below.

*TF-binding-site insertion analysis*. A total of 13 TSRs was selected from the human genome (Supplementary Table 2) and DNA inserts were designed to introduce 7 TF binding sites at positions −50, −20 and +25 bp relative to the primary TSS (Sp1, NRF1, NFY, YY1, p53, CTCF or a control sequence; Fig. 3b,c). Binding-site insertion replaced the endogenous sequence at each location to maintain the relative spacing of regulatory element DNA outside of the TF binding site insertion. For RNA samples, the transcriptional output of each sequence was summarized by counting alignments with start sites near the designated promoter region (±7 bp from the TSS). These values were scaled based on plasmid DNA levels and transformed as described above. Overall levels were reported as the $\log_2$-transformed ratio relative to the control binding site insertion that does not match any known TF binding sites.

*TF-binding-site sweep analysis*. To unbiasedly assess the position-dependent impact of TF binding sites on transcription initiation levels, we first designed a synthetic promoter sequence that lacked matches to any of the known motifs in the HOMER TF motif library (Supplementary Table 2). TF binding sites corresponding to Sp1, NFY, NRF1 and YY1 were then inserted at 2 bp intervals along the length of the sequence (Fig. 4a–c). NRF1 was additionally inserted at the same 2 bp intervals into either the WT TOB2 enhancer or a mutant version lacking endogenous NRF1 binding sites (Fig. 4e). Binding-site insertion replaced the endogenous sequence at each location to maintain the relative spacing of regulatory element DNA outside of the TF binding site insertion. To analyse the impact of each TF binding site sweep, a scaling factor for each insert sequence was determined by calculating min(10,000/plasmid, 100). After multiplying the start-site counts by the scaling factor, a pseudocount of 1 was added and the values were $\log_2$ transformed. Replicates were then merged by averaging. Reference coverage was defined for each promoter–TF binding site pair by calculating the average position-wise signal across all of the sequences (Extended Data Fig. 7). The difference between the merged signal and reference coverage was smoothed for visualization using the R loess function with span parameter set to 0.1.

*TF-binding-site mutation analysis*. In total, 133 TSRs were randomly selected and TF binding sites matching 20 different motifs were mutated to monitor their impact on transcription initiation patterns. In addition to the wild-type TSR sequence [−113, +42], a separate insert was designed for each motif where one or more TF binding sites were found. Sequences corresponding to the TF binding site were replaced with the same control sequence in each case starting at the 5′ end and continuing the length of the binding site (control sequence: TAACTGTAATAC-CTCCTGAAGTC). Only motifs with matches to at least 20 different TSRs were used in the analysis (Sp1, NFY, NRF1, YY1 and ETS; Fig. 5e and Extended Data Fig. 10d). DNA-scaled and log-transformed start site values were calculated as with the above site sweep analysis. For each motif, start positions were classified as enriched or depleted based on an overrepresentation analysis in genomic contexts ($P_{adj} < 0.01$,

data from U2OS motif enrichment analysis; Fig. 1d). TSS shifts were determined by finding the weighted mean TSS position per insert, then subtracting the mean position per mutant from the mean position of the relevant control.

*Reproducibility analysis*. The reproducibility of MPRA results was assessed by comparing (1) the variation in initiation activity levels among different barcode replicates for the four TSRs displayed in Fig. 3b (Extended Data Fig. 6a); (2) comparing summary heat maps of the TSSs and their normalized activity levels captured by TSS-MPRA for a 2 bp incremental sweep of TF binding site sweeps (Sp1, NRF1, NFY and YY1) from −100 to +40 using four different barcode sets (Extended Data Fig. 7a–e); and (3) comparing TSS activity levels for a given DNA fragment across at least two biological replicates and between independent barcodes for each TSS-MPRA library (Supplementary Figs. 6 and 7).

## Reporting summary

Further information on research design is available in the Nature Portfolio Reporting Summary linked to this article.

## Data availability

All raw and processed data generated for this study can be accessed under NCBI GEO accession number GSE199431. Previously published GEO and high-throughput sequencing datasets analysed as part of this study include csRNA-seq data in C57BL/6 mouse macrophages (GSE135498), *NFY*-knockdown Start-seq data in mouse MEFs (GSE115110), PRO-cap data from 69 human LCLs (GSE110638), NRF1 ChIP–seq data from ENCODE in HepG2 (ENCSR853ADA) and K562 (ENCSR494TDU) cells. Genomes used for the analysis of sequencing data include human: GRCh38/hg38 (https://hgdownload.soe.ucsc.edu/goldenPath/hg38/bigZips/hg38.fa.gz); mouse (C57BL/6): GRCm38/mm10 (https://hgdownload.soe.ucsc.edu/goldenPath/mm10/bigZips/mm10.fa.gz); mouse (SPRET): GCA_001624865.1 (https://www.ncbi.nlm.nih.gov/datasets/genome/GCA_001624865.1/); and green monkey: *Chlorocebus sabeus* 1.1/chlSab2 (https://hgdownload.soe.ucsc.edu/goldenPath/chlSab2/bigZips/chlSab2.fa.gz). Gene annotations were downloaded from GENCODE (human v34, mouse v25; https://www.gencodegenes.org/). Disease-risk variants from the GWAS Catalog mapping to hg38 were downloaded from the UCSC Genome Browser (https://hgdownload.soe.ucsc.edu/goldenPath/hg38/database/gwasCatalog.txt.gz). Liftover files for mapping between mouse strains were obtained online (http://hgdownload.cse.ucsc.edu/goldenpath/mm10/liftOver/mm10ToGCA_001624865.1_SPRET_EiJ_v1.over.chain.gz (C57BL/6/mm10 to SPRET) and https://hgdownload.soe.ucsc.edu/goldenPath/GCA_001624865.1_SPRET_EiJ_v1/liftOver/GCA_001624865.1_SPRET_EiJ_v1ToMm10.over.chain.gz (SPRET to C57BL/6/mm10)). Per-chromosome VCF files containing genotyping data for the samples analysed previously[63] were downloaded from the 1000 Genomes Project (ftp://ftp.1000genomes.ebi.ac.uk/vol1/ftp/data_collections/1000G_2504_high_coverage/working/20201028_3202_raw_GT_with_annot/). Source data are provided with this paper.

## Code availability

Code used to analyse data in this Article has been integrated into HOMER, or is available from the following repositories as described in the methods: HOMER2 (HOMER v.5) (http://homer.ucsd.edu/homer2/), MEIRLOP v.0.0.16 (https://github.com/npdeloss/meirlop) and MEPP v.0.0.1 (https://github.com/npdeloss/mepp).

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

**Acknowledgements** We thank C. D. A. Saldanha for expert technical assistance and L. V. Ngoc, J. T. Kadonaga and the members of the Duttke laboratory for discussions of the manuscript; and the staff at Life Science Editors for editing services. This work was supported by NIH grants R00GM135515 and WSU startup funds (to S.H.D.); R01GM134366, U01DA051972, U01AI150748, U19AI135972, P01HL147835 and R01MH127077 (to C.B.); and R01GM129523 (to S.H.). S.H. received additional support from NIH grants R01GM134366, P30DK063491, P30DK120515, U01AI150748, U19AI135972 and R21DA056177. A.F.C. was supported by NIH career award K08AI130381 as well as a Career Award for Medical Scientists from the Burroughs Wellcome Fund; C.G. in part by predoctoral fellowship F31HG011823; N.P.D.S. in part by an NLM Training Grant (T15LM011271) and the Katzin Prize Endowed Fund. B.R.M. is a WSU STARS scholar. This publication includes data generated at the UC San Diego IGM Genomics Centre using an Illumina NovaSeq 6000 system that was purchased with funding from a National Institutes of Health SIG grant (S10 OD026929).

**Author contributions** S.H.D., A.F.C., S.H. and C.B. oversaw the overall design and execution of the project. The experiments were performed by S.H.D., J.X., C.G. and S.H. The computational analyses were performed by S.H.D., M.C., N.P.D.S., B.R.M. and C.B.; S.H.D. and C.B. were primarily responsible for writing the manuscript.

**Competing interests** The authors declare no competing interests.

**Additional information**
**Correspondence and requests for materials** should be addressed to Sascha H. Duttke, Sven Heinz or Christopher Benner.

Most current motif
analysis methods
(HOMER, MEME, etc.)
normalize GC%

## Fragment GC content

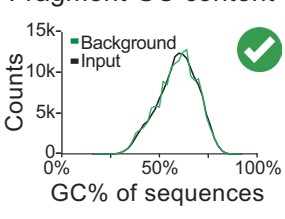

## Positional (single-nucleotide) GC bias

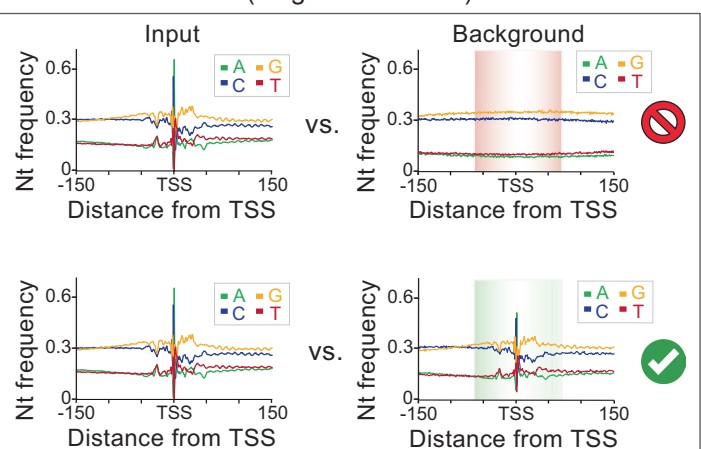

**HOMER2**
normalizes GC%
and positional biases,
e.g., nt bias of TSS
or mutation likelyhoods

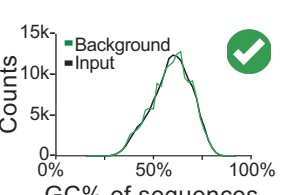

*While in this example figure uses TSSs, any biological or user-defined positional information can be integrated.*

**Extended Data Fig. 1 | HOMER2 - A new TF motif and sequence analysis approach that allows controlling for both single-nucleotide positional and fragment-wide sequence biases.** By contrast to most current motif finding methods that normalize across the complete sequence fragment in the analysis, HOMER2 accounts for both fragment-wide and single-nucleotide positional biases of input sequences when it selects background sequences from the genome, such as nucleotide preferences naturally found near TSSs.

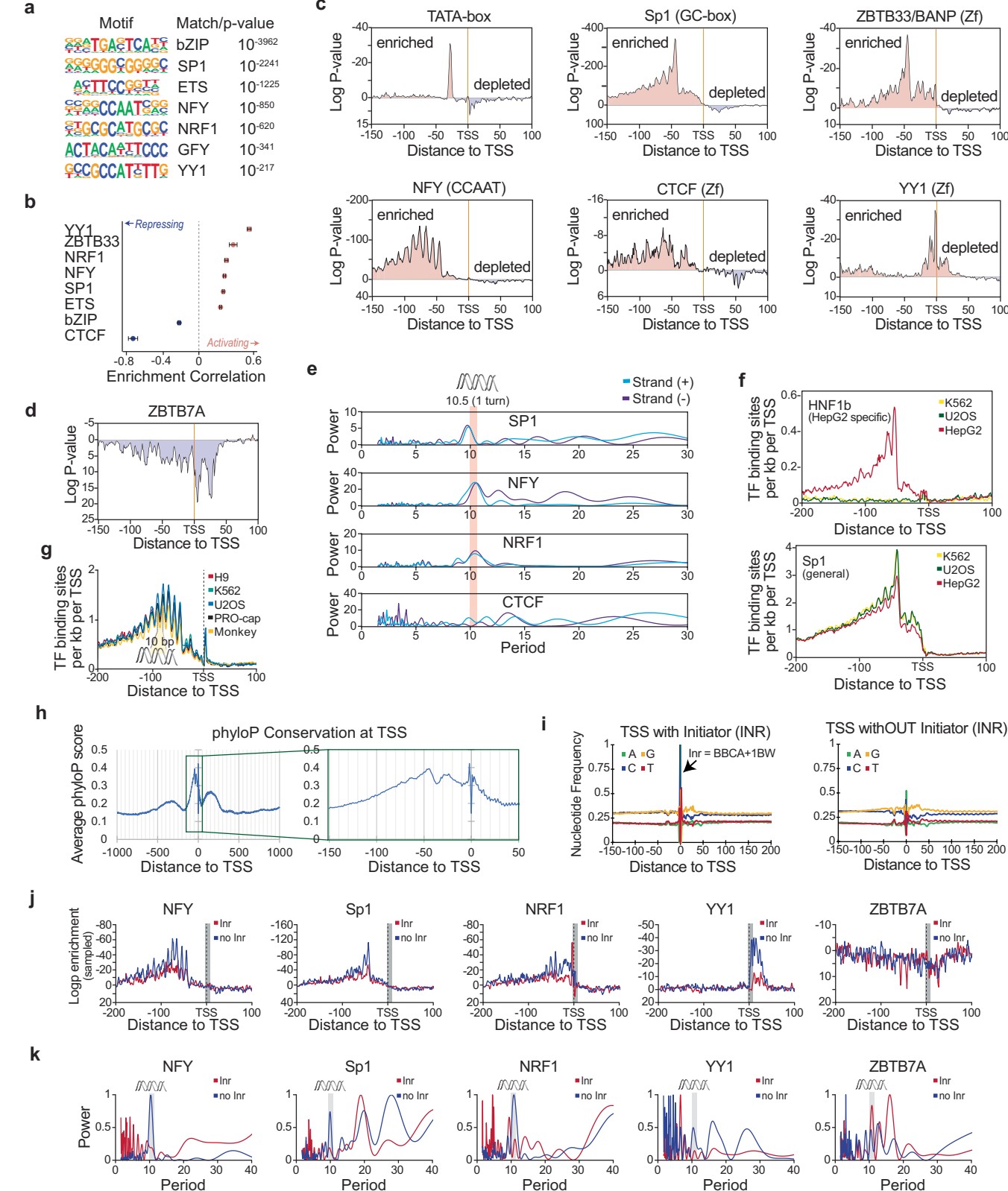

**Extended Data Fig. 2** | See next page for caption.

**Extended Data Fig. 2 | TSS-centric analysis reveals a spatial grammar of TFs.**
**a**, De novo motif enrichment analysis of TSRs active in U2OS cells by HOMER2 reveals the TF motifs with the highest enrichment in transcribed regulatory elements. **b**, Association of TSR-enriched TF binding sites with transcription initiation frequency calculated using MEIRLOP[75] using initiation strength as covariant. **c**, Examples of TF binding sites with natural preferential positioning in the proximity of human TSSs. Positional enrichment or depletion was calculated using HOMER2, accounting for both positional (i.e. TSS-proximal), and fragment-wide nucleotide content bias. **d**, Binding sites of the repressor ZBTB7A are depleted near active TSS, especially downstream where the RNA Polymerase II initiation complex is proposed to initially contact the TSR. **e**, Many TF binding sites including Sp1, NFY, and NRF1, but not all (i.e., CTCF) have preferred 10.5 bp helical positioning relative to active TSS when found between −120 and −40 bp, as shown by Fourier analysis (please see Methods for details). **f**, Binding sites of cell type-specific activator TFs often show preferential positioning relative to the TSS only in cells that expressed them. TF binding site distribution profiles for HepG2-specific HNF1 and ubiquitous Sp1/GC-box motifs across TSSs identified in K562, U2OS and HepG2 cell lines by csRNA-seq.

**g**, Preferential TF binding site localisation is highly conserved across species and methods. Motif density plot of the NFY binding site relative to TSS identified using csRNA-seq from different human and green monkey (Vero cells) cell lines as well TSS identified using PRO-cap in K562 cells[86]. **h**, The upstream, rather than the downstream promoter region is more conserved. Aggregate PhyloP scores at single base resolution relative to active TSSs in U2OS cells reveals that upstream regions, especially around −30 bp and −50 bp, relative to the TSS, are preferentially conserved. **i**, Genomic nucleotide frequency plots relative to TSS containing or lacking a canonical Initiator motif (BBCA + 1BW)[36] at the TSSs. **j**, Frequency and patterns of position-specific TF binding sites are more eminent relative to TSSs that lack canonical core promoter motifs. Normalized NFY, Sp1, NRF1, YY1 and ZBTB7A motif occurrences are displayed relative to the TSS containing (red) or lacking (blue) a canonical human Initiator motif (BBCABW) at the TSS (grey). **k**, Helical periodicity of TF binding sites found between −120 to −40 bp relative to the TSS are more prominent in TSS lacking a canonical human Initiator motif (BBCABW). Fourier analysis of TF binding sites NFY, Sp1 NRF1, YY1 and ZBTB7A revealed preferred 10.5 bp helical positioning relative to TSS lacking a human Initiator in position-dependent TF factors.

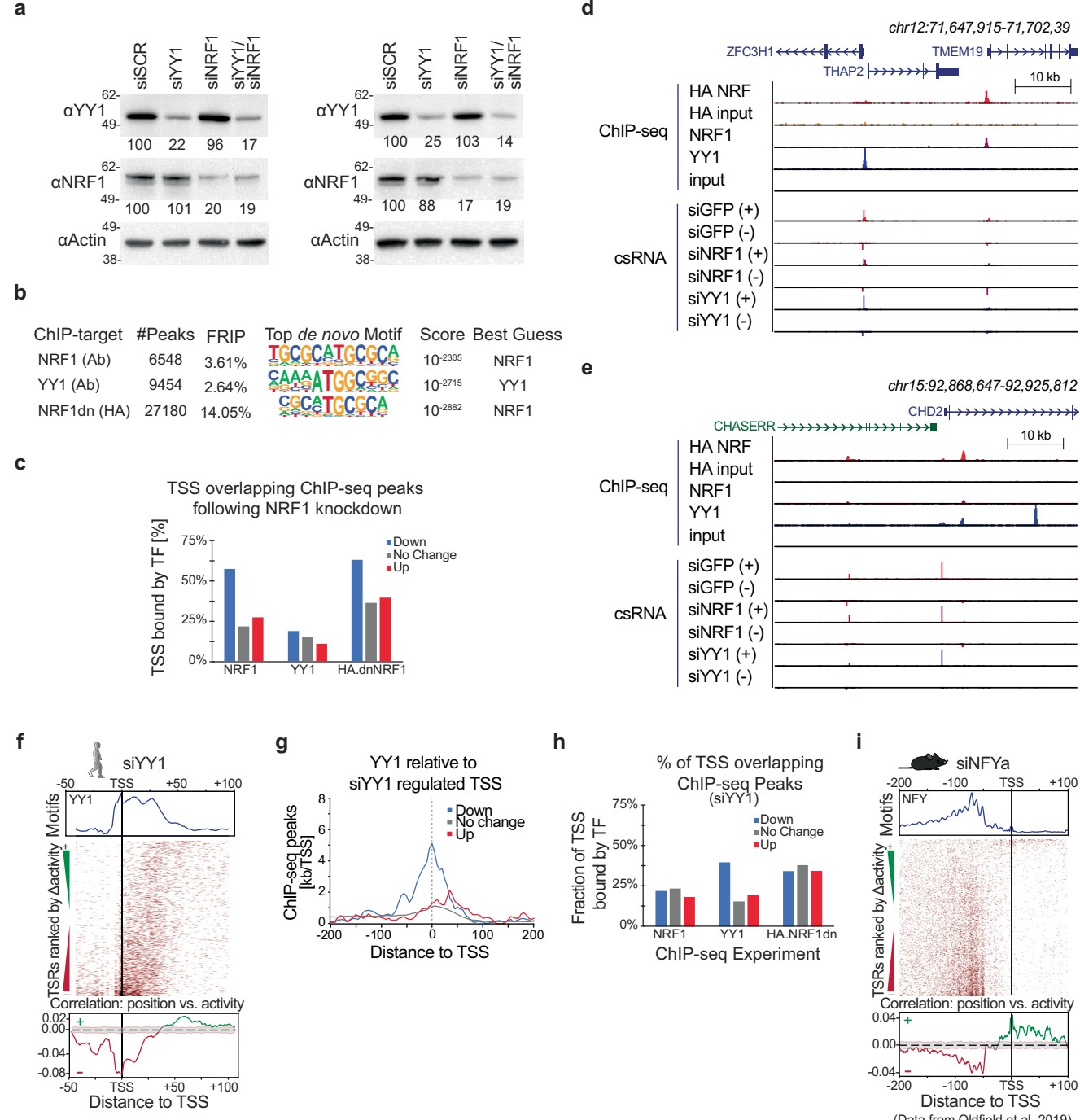

**Extended Data Fig. 3 | TF occupancy at differentially positioned binding sites. a**, Quantification of TF knockdown: Western blot 24 h following knockdown of YY1 and NRF1 in replicates using beta-Actin as a control (n = 2, representative experiment shown). For original images please see Supplementary Fig. 4. **b**, Validation of ChIP-seq data: De novo motif finding of ChIP-seq peaks using HOMER2 identifies the expected motif for each antibody target. FRIP stands for Fraction of Reads In Peaks. **c**, Overlap between NRF1, YY1, and dnNRF1 binding and TSS reveals enhanced binding of NRF1 to TSSs both up and down regulated by siRNA targeting NRF1 relative to invariant TSS. **d,e**, Example loci with ChIP and csRNA-seq data. **f**, Position dependent function of human YY1.

Human U2OS TSSs were ranked from gain to loss of transcription initiation activity upon YY1 knockdown and analysed for YY1 motif positional enrichment (dark red). **g**, TSSs downregulated upon YY1 knockdown have YY1 bound within its preferred region, as assessed by ChIP-seq, while derepressed TSSs have YY1 binding further downstream. **h**, Overlap between NRF1, YY1, and dnNRF1 binding and TSS reveals enhanced binding of YY1 to TSSs both up and down regulated by siRNA targeting YY1 relative to invariant TSS. **i**, Position dependent function of mouse NFY. Mouse embryonic fibroblast (MEF) TSSs were ranked from gain to loss of transcription initiation activity upon NFYa knockdown[39] and analysed for NFY motif positional enrichment.

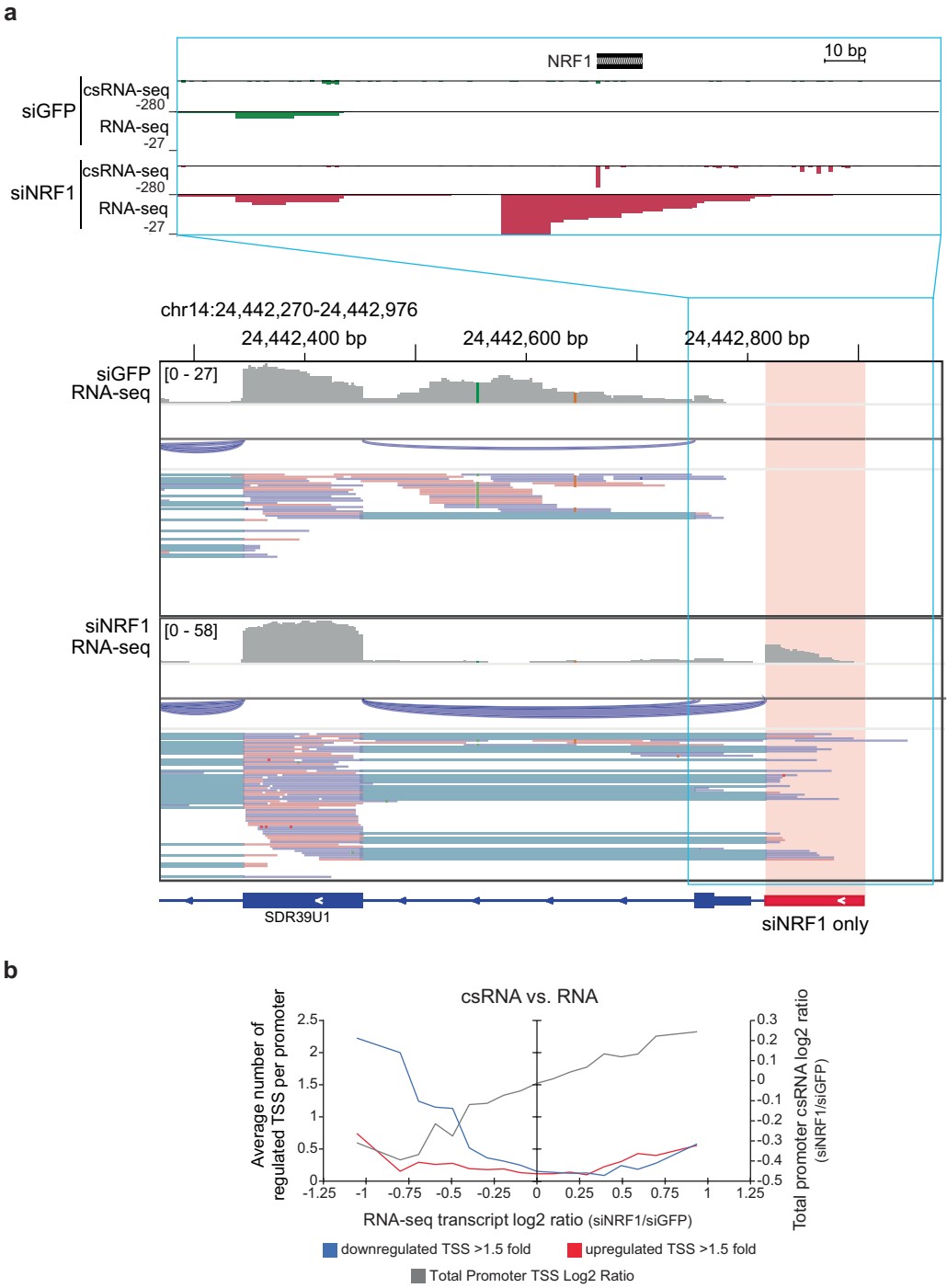

**Extended Data Fig. 4 | Differential TSS usage can impact gene isoform usage and gene expression. a**, Example locus (*SDR39U1*) where loss of NRF1 by siRNA knockdown led to the induction of several TSSs near to and upstream of a NRF1 binding site motif (top). RNA-seq profiling revealed that cells treated with NRF1 siRNA expressed a novel isoform with unique splice junctions not observed in the control sample (bottom). **b**, Changes in TSSs levels impact gene expression. Moving average of the number of either upregulated or downregulated TSS overlapping the annotated promoter (within 200 bp) of genes ranked by their change in RNA-seq transcript levels (orange, grey). Also depicted is the average of the total promoter csRNA-seq level change (i.e. integrated across all TSS in the promoter region, blue).

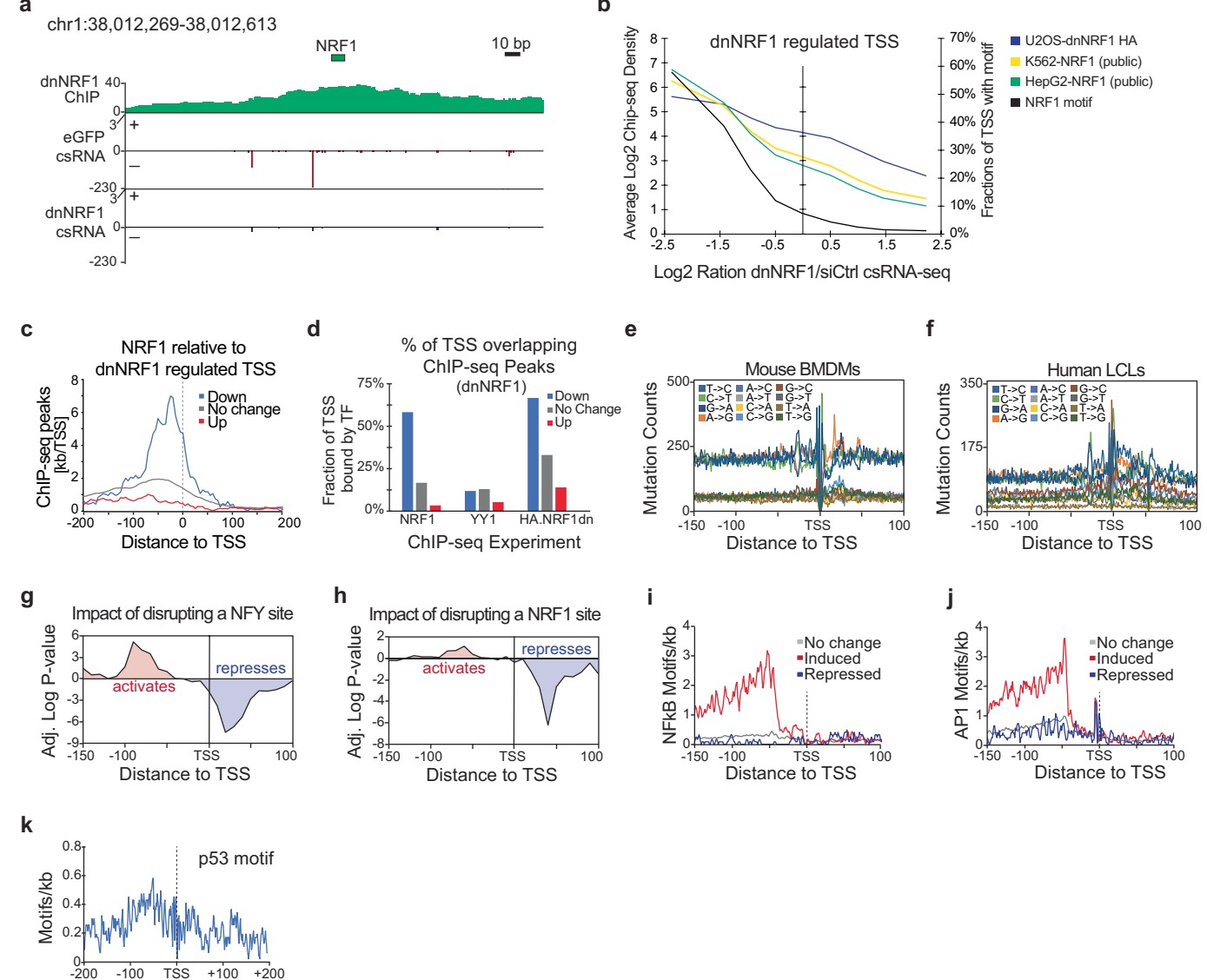

**Extended Data Fig. 5 | dnNRF and natural genetic variation analysis.**
Overexpression of dnNRF1 results in repression of transcription initiation at TSRs in the vicinity of dnNRF1 binding sites. **a**, Genome browser tracks at an example locus (*UTP11*) showing the HA-tagged dnNRF1 ChIP-seq read density and normalized csRNA-seq TSS activity levels in eGFP or dnNRF1 expressing U2OS cells. **b**, TSRs strongly down-regulated by dnNRF1 expression are also bound by dnNRF1, as assessed by ChIP-seq. The average ChIP-seq normalized read density or fraction of TSRs containing the NRF1 binding site from −150 to +50 relative to the TSS are plotted as a function of the log2 ratio of TSS activity between dnNRF1 and GFP expressing U2OS cells as measured by csRNA-seq. **c**, TSSs downregulated upon overexpression of HA-tagged dominant negative NRF1 (dnNRF1) knockdown have NRF bound within its preferred region upstream of the TSS, as assessed by ChIP-seq. **d**, Overlap between NRF1, YY1, and dnNRF1 binding and TSS reveals enhanced binding of NRF1/dnNRF1 to TSSs down regulated by dnNRF1 expression relative to invariant TSS. **e**, Distribution of single nucleotide variants relative to the TSS used in the analysis of mouse (C57Bl/6 and SPRET) bone marrow derived macrophages (BMDMs) comparing

different strains and **f**, human tssQTLs found in LCLs. **g**, Analysis of the genome-wide significance of the association between mutations in the NFY binding site, or **h**, NRF1 binding site and the change in transcription initiation in macrophages from each mouse strain, calculated for each TF binding site as a function of their relative distance to the TSS. Positive logP values indicate that mutations predicted to cause reduced TF binding are more strongly associated with reduced initiation, while negative logP values indicate that the mutated TF binding sites are more strongly associated with increased initiation. Distance-dependent profiles were calculated using TF binding sites identified in overlapping windows of 30 bp at 10 bp increments from −150 to +100 bp relative to the TSS. **i**,**j**, TF binding sites for TLR4 pathway activated TFs that recruit RNA polymerase II are preferentially positioned relative to TSSs that increase transcription following KLA treatment. Motif distribution profiles relative to TSSs of TSRs that were induced, repressed or did not change upon stimulation of bone marrow derived macrophages with KLA for the binding sites of **i**, NF-κB (p65) and **j**, AP1. **k**, Distribution of the p53 DNA binding site relative to active TSS from U2OS cells.

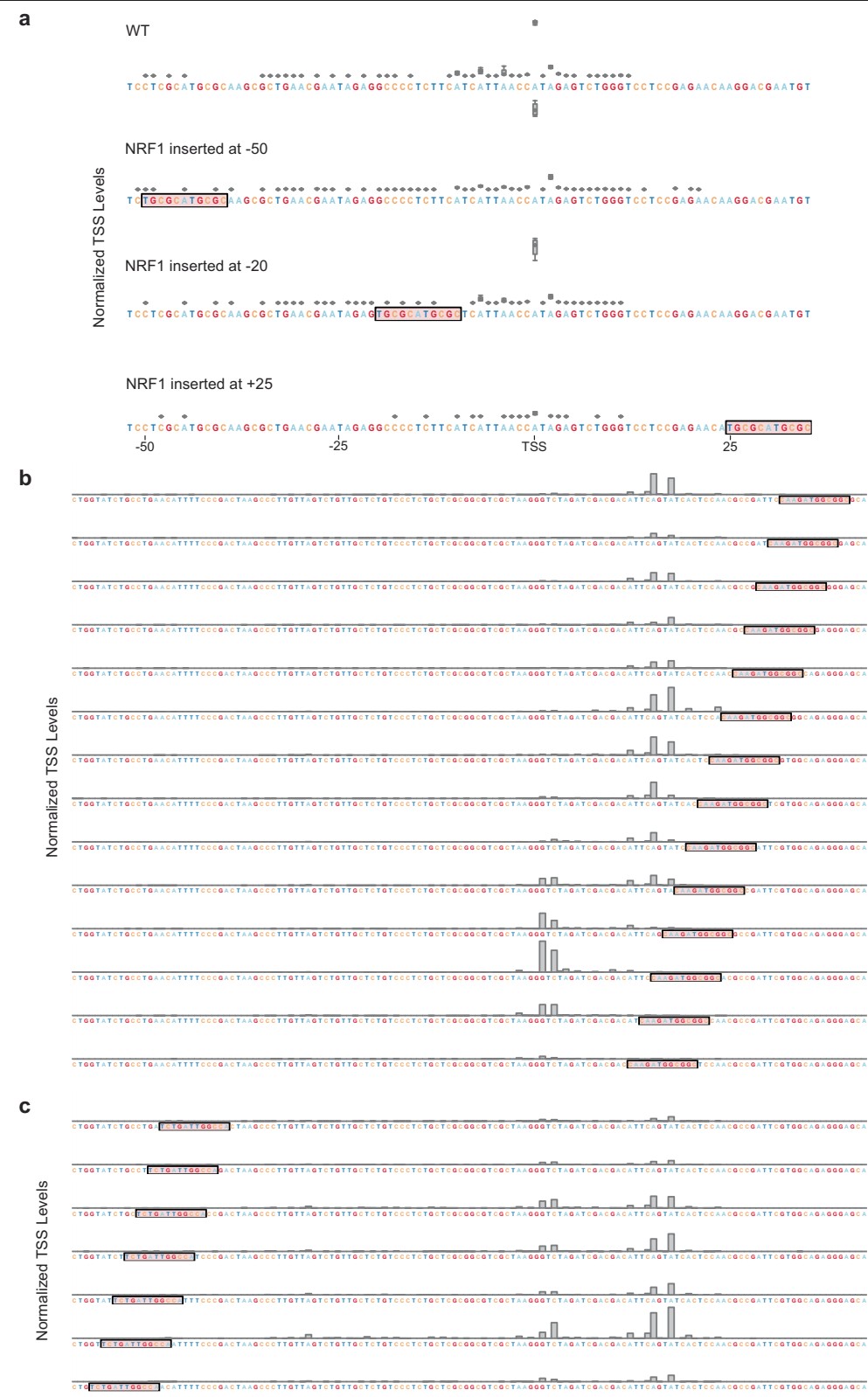

**Extended Data Fig. 6 | TSS-MPRA results are highly reproducible. a**, Variation in initiation activity levels among different barcode replicates for the four TSRs displayed in Fig. 3b that shows the impact of differential NRF1 binding site position on TSS activity for a TSR from the *EIF2S1* locus (depicted in sense). TSS-MPRA captures the impact of adjusting TF binding site positions on transcription initiation at single-nucleotide resolution. **b**, Normalized TSS activity profiles on a synthetic DNA insert measuring the impact of adjusting the YY1 binding site position by 2 bp increments, showing waves of increased and reduced transcription initiation and shifting TSS. **c**, Examples of normalized TSS activity profiles measured by adjusting the position of the NFY binding site every 2 bp, showing the importance of helical positioning for TF potency in recruiting RNAP II.

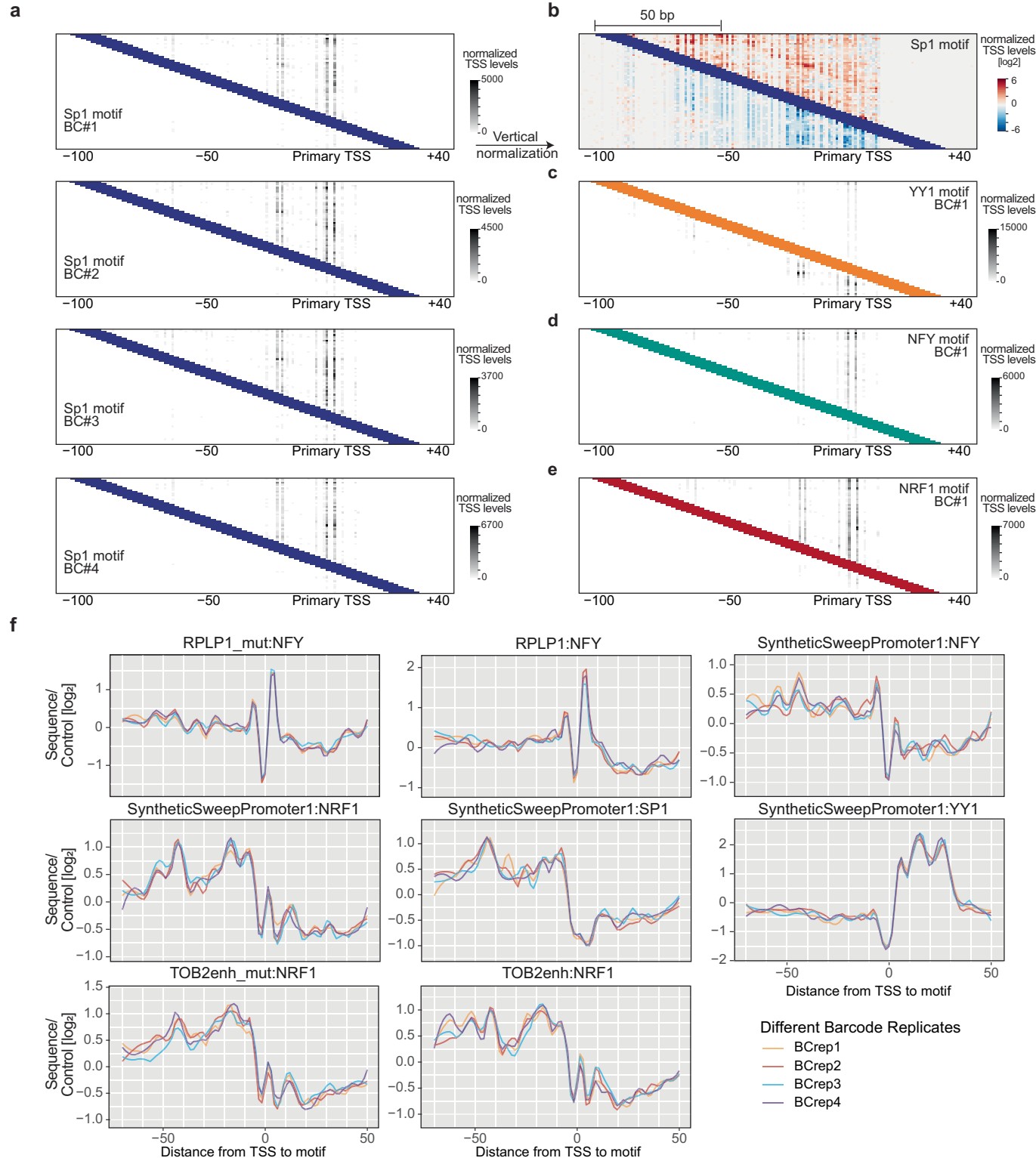

**Extended Data Fig. 7 | TSS-MPRA analysis of TF binding site sweeps reveal additional evidence for position-dependent TF function. a**, Summary heat maps of the TSSs and their normalized activity levels captured by TSS-MPRA for a 2 bp incremental sweep of the Sp1 binding site sweep from −100 to +40 across an artificial, TF motif-depleted DNA background with four different barcode sets. The Sp1 binding site position is shown in blue. **b**, Vertical normalization of the Sp1 binding site sweep TSS-MPRA reports the log2 fold change in TSS activity relative to the average activity of that TSS across all possible Sp1 binding site positions. This normalization highlights TSSs that are activated (red) and repressed (blue) relative to the average level of activity for

each binding site position. The Sp1 binding site position is shown in blue. **c**, Summary heat maps of the TSSs and activity levels captured by TSS-MPRA for a 2 bp sweeps of the YY1 binding site, **d**, NFY binding site and **e**, NRF1 binding site, sweep from −100 to +40 across an artificial, TF motif-depleted DNA background. Only BC#1 of 4 is shown. **f**, Lineplots showing that the position-dependent impact of sweeping TF binding sites in a synthetic sequence is highly reproducible as independently assessed for each of the four barcodes sets. Data reported in the manuscript were obtained by averaging all four barcodes and both biological replicates.

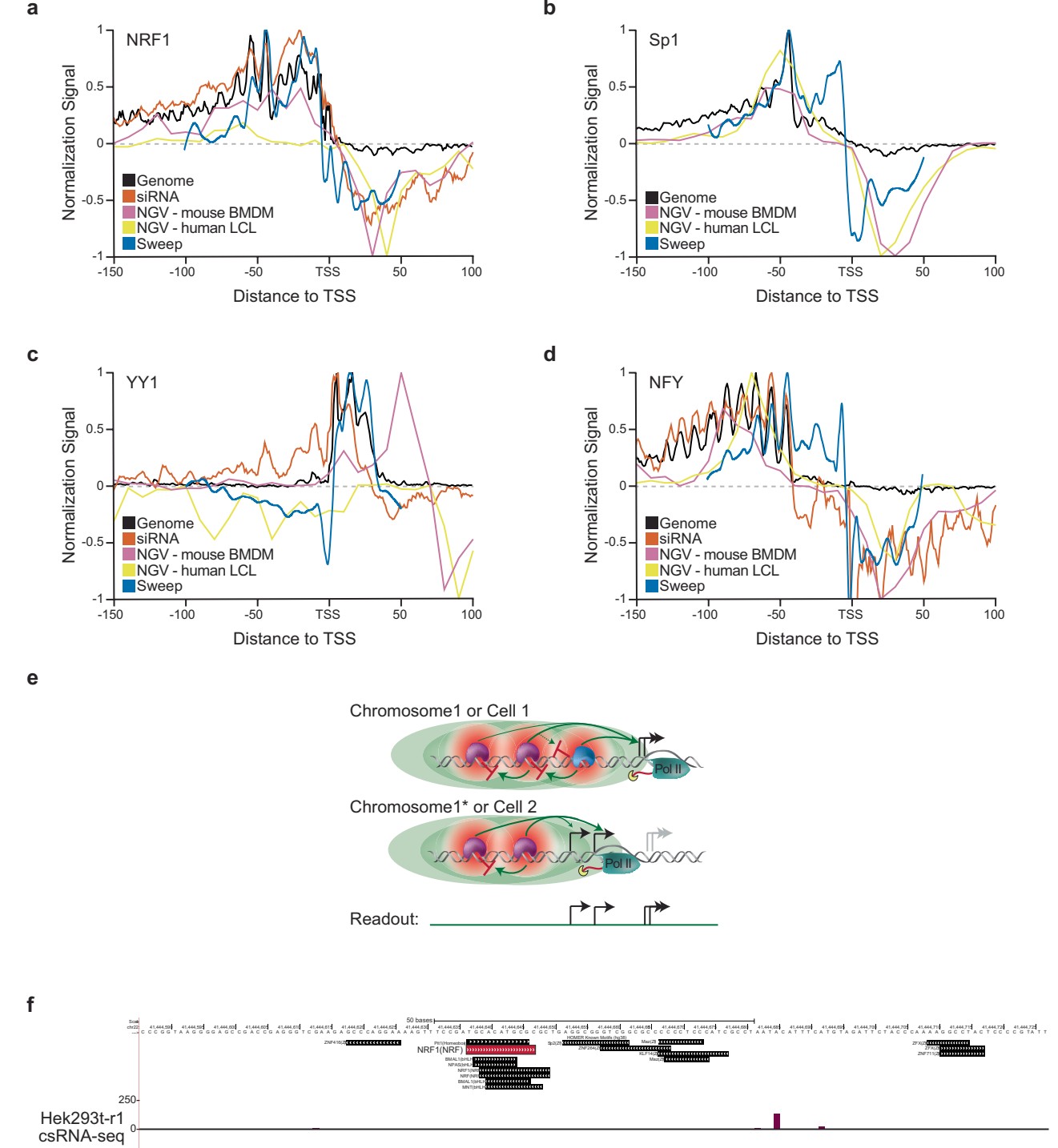

**Extended Data Fig. 8 | Multiple experimental approaches reveal consistent position-dependent functional profiles that are unique to each TF.** Comparison of patterns from natural preferred TF binding site positional enrichment in the genome relative to active TSSs (black, i.e. Fig. 1d), impact of TF knockdown on transcription of proximate TSSs as a function of distance to the TF binding site (orange, i.e. Fig. 1e, flipped), impact of TF binding site mutations due to natural genetic variation on transcription (pink/yellow, i.e. Figs. 2f, 5a) and a binding site's ability to impact transcription as captured by TF binding site sweeps with TSS-MPRA (blue, i.e. Fig. 4c) altogether reveal consistent, position-dependent functions and superhelical preferences for

**a**, YY1, **b**, Sp1, **c**, NRF1 and **d**, NFY. Each profile was scaled such that the most extreme value was set to 1/−1. **e**, Hypothetical model for TF-mediated TSS selection and dispersed initiation. TFs can recruit or block transcription initiation based on their spacing. In most TSRs, this spacing-dependent function of TFs is integrated over several TFs. As TF binding is transient, different sets of TFs can be present at a given moment at homologous TSRs in sister chromosomes or different cells of the same kind or vary at the same TSR over time. **f**, The transcribed putative *TOP2* enhancer region contains an NRF1 binding site. UCSC browser image and HOMER-annotated motifs with the NRF1 binding site mutated in the screen highlighted in red.

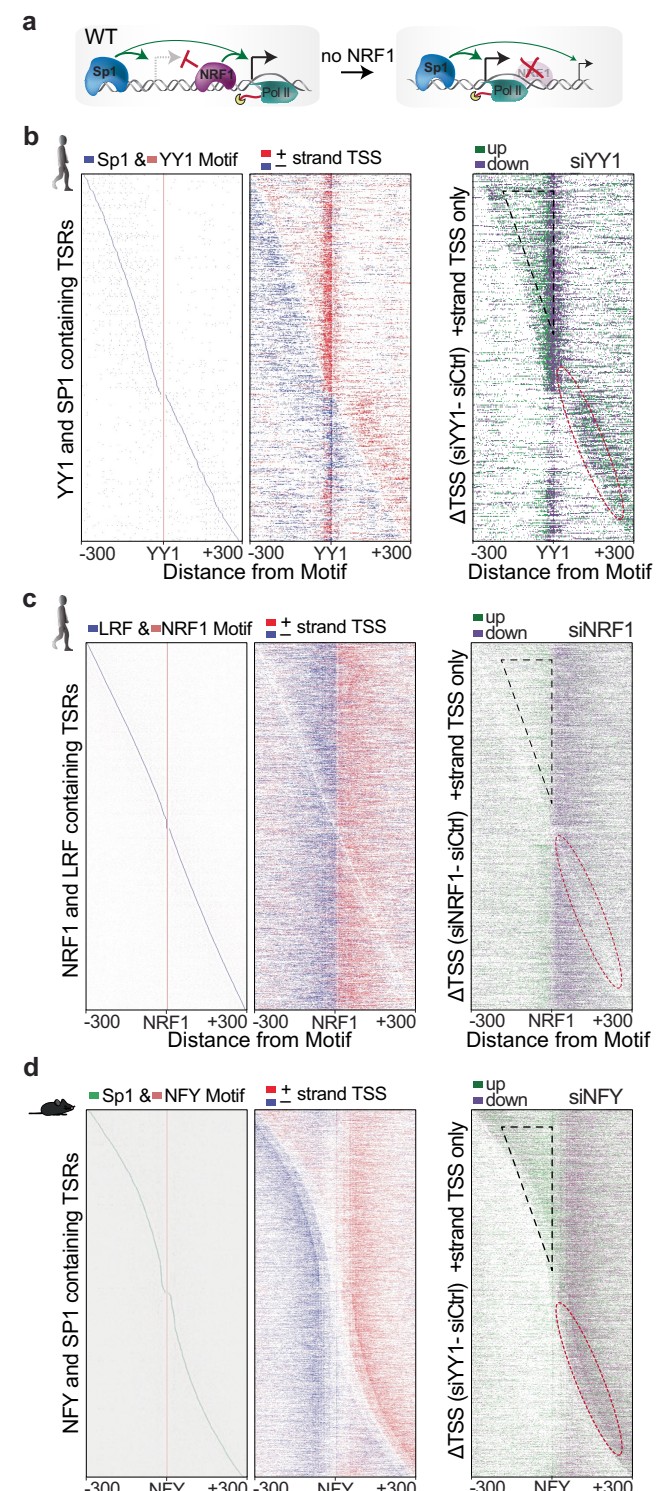

**Extended Data Fig. 9 | Spacing between TFs can coordinately guide transcription initiation. Additional examples of TF-TF interaction. a**, Model for TF-mediated RNA Polymerase II initiation and coordinated TSS selection by activator TFs NRF1 and Sp1 based on their spatial preferences. TSRs containing both **b**, YY1 and Sp1 binding sites, **c**, NRF1 and ZBTB7A (LRF) binding sites, and **d**, NFY and Sp1 binding sites, sorted by the distance between the TF binding sites with csRNA-seq initiation levels shown in forward (red) and reverse (blue) direction. The impact of YY1, NRF1, and NFY siRNA knockdown on activity for + strand TSSs are shown on the right with upregulated TSSs shown in green and downregulated TSSs in purple. TSS patterns and their regulation at YY1 and Sp1 binding sites containing loci reflect the unique preferred initiation profiles associated with the YY1 and Sp1 binding sites (**b**), while TSS patterns between the ZBTB7A and NRF1 binding sites show little to no interaction (**c**). **d**, Analysis of the Sp1 and NFY in mouse fibroblasts[39] suggests conservation of position-dependent collaborative TF function across mammals.

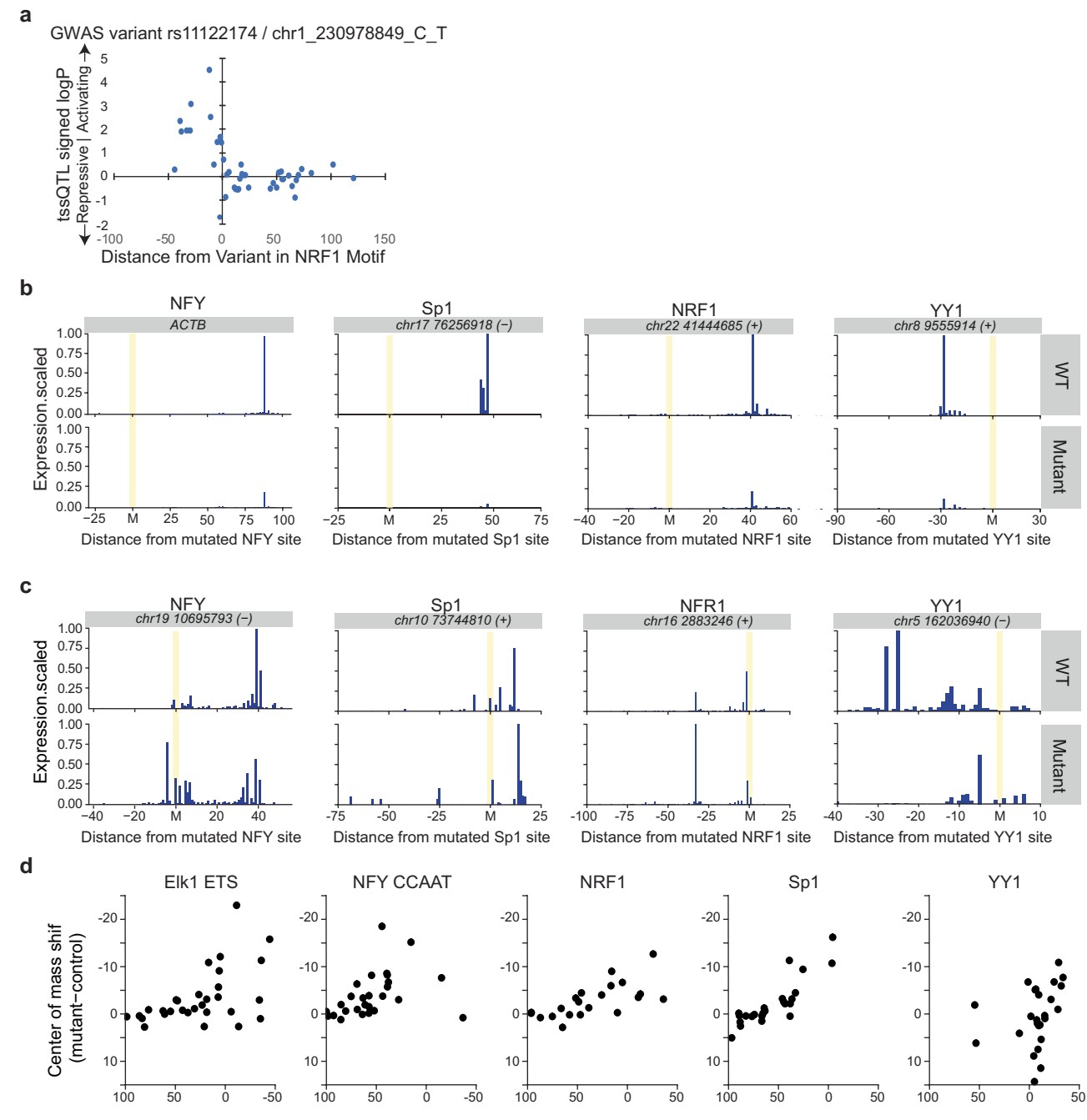

**Extended Data Fig. 10 | Position-dependent TF function in human health and disease. a**, Disease-associated variant rs11122174, identified through GWAS, is found within an NRF1 binding site and displays position-dependent changes in tssQTL significance relative to nearby TSS. **b**,**c**, Massively parallel mutation analysis of human regulatory elements reveals position-dependent TF function. Mutations of preferentially positioned TF binding sites result in loss of transcriptional activity (**b**), while mutation of TF binding sites in vicinity to TSSs lead to ectopic TSSs (**c**, derepression), demonstrating the dual, position-dependent function of NFY, Sp1, NRF1, and YY1 in human regulatory elements. Mutation of TSS-proximal TF binding sites was also associated with notable changes in TSS selection and thus alternative 5′UTRs, a hallmark of many diseases. **d**, Relationship of TF binding site position and impact on TSS selection: Mutation of TF binding sites near TSS or within their naturally enriched positions had the strongest effect on the TSS pattern of regulatory elements while those outside thereof, had less impact.

Duttke, Sascha
Heinz, Sven

# Reporting Summary

## Statistics

For all statistical analyses, confirm that the following items are present in the figure legend, table legend, main text, or Methods section.

| n/a | Confirmed | |
|---|---|---|
| ☐ | ☒ | The exact sample size (*n*) for each experimental group/condition, given as a discrete number and unit of measurement |
| ☐ | ☒ | A statement on whether measurements were taken from distinct samples or whether the same sample was measured repeatedly |
| ☐ | ☒ | The statistical test(s) used AND whether they are one- or two-sided<br>*Only common tests should be described solely by name; describe more complex techniques in the Methods section.* |
| ☐ | ☒ | A description of all covariates tested |
| ☐ | ☒ | A description of any assumptions or corrections, such as tests of normality and adjustment for multiple comparisons |
| ☐ | ☒ | A full description of the statistical parameters including central tendency (e.g. means) or other basic estimates (e.g. regression coefficient) AND variation (e.g. standard deviation) or associated estimates of uncertainty (e.g. confidence intervals) |
| ☐ | ☒ | For null hypothesis testing, the test statistic (e.g. *F*, *t*, *r*) with confidence intervals, effect sizes, degrees of freedom and *P* value noted<br>*Give P values as exact values whenever suitable.* |
| ☒ | ☐ | For Bayesian analysis, information on the choice of priors and Markov chain Monte Carlo settings |
| ☒ | ☐ | For hierarchical and complex designs, identification of the appropriate level for tests and full reporting of outcomes |
| ☐ | ☒ | Estimates of effect sizes (e.g. Cohen's *d*, Pearson's *r*), indicating how they were calculated |

*Our web collection on statistics for biologists contains articles on many of the points above.*

## Software and code

Policy information about availability of computer code

| Data collection | No specialized software was used for data collection |
|---|---|

| Data analysis | Code used to analyze data in this manuscript has been integrated into HOMER, or is available from the following repositories as described in the methods:<br>HOMER2 (HOMER v5) (http://homer.ucsd.edu/homer2)<br>MEIRLOP v0.0.16 (https://github.com/npdeloss/meirlop)<br>MEPP v0.0.1 (https://github.com/npdeloss/mepp)<br>MAGGIE v1.1.1 (https://github.com/zeyang-shen/maggie)<br><br>Additional Software:<br>cutadapt (v3.4)<br>STAR (v2.7.10a)<br>R (v4.2.2)<br>DESeq2 (v1.38.3)<br>bcftools (v1.6)<br>plink2 (v2.00a2.3LM)<br>TensorQTL (v1.0.3)<br>Fiji (V 1.53j)<br>Cluster 3.0 (v3.0)<br>Java TreeView (v1.1.6r4)<br>Excel (v16.83)<br>UCSC and IGV online Genome Browsers. |
|---|---|

For manuscripts utilizing custom algorithms or software that are central to the research but not yet described in published literature, software must be made available to editors and reviewers. We strongly encourage code deposition in a community repository (e.g. GitHub). See the Nature Portfolio guidelines for submitting code & software for further information.

# Data

Policy information about availability of data

All manuscripts must include a data availability statement. This statement should provide the following information, where applicable:
- Accession codes, unique identifiers, or web links for publicly available datasets
- A description of any restrictions on data availability
- For clinical datasets or third party data, please ensure that the statement adheres to our policy

All raw and processed data generated for this study can be accessed at NCBI Gene Expression Omnibus (GEO; https://www.ncbi.nlm.nih.gov/geo/) accession number GSE199431 (https://www.ncbi.nlm.nih.gov/geo/query/acc.cgi?acc=GSE199431). Previously published GEO and high-throughput sequencing datasets analyzed as part of this study include csRNA-seq data in C57Bl/6 mouse macrophages (GSE135498, https://www.ncbi.nlm.nih.gov/geo/query/acc.cgi?acc=GSE135498), NFY knockdown Start-seq data in mouse MEFs (GSE115110, https://www.ncbi.nlm.nih.gov/geo/query/acc.cgi?acc=GSE115110), PRO-cap data from 69 human lymphoblastoid cell lines (GSE110638, https://www.ncbi.nlm.nih.gov/geo/query/acc.cgi?acc=GSE110638), NRF1 ChIP-seq data from ENCODE in HepG2 (ENCSR853ADA, https://doi.org/doi:10.17989%2FENCSR853ADA) and K562 (ENCSR494TDU, https://doi.org/doi:10.17989%2FENCSR494TDU) cells. Genomes used for the analysis of sequencing data include Human: GRCh38/hg38 (https://hgdownload.soe.ucsc.edu/goldenPath/hg38/bigZips/hg38.fa.gz), Mouse(C57Bl/6): GRCm38/mm10 (https://hgdownload.soe.ucsc.edu/goldenPath/mm10/bigZips/mm10.fa.gz), Mouse(SPRET): GCA_001624865.1 (https://www.ncbi.nlm.nih.gov/datasets/genome/GCA_001624865.1/), and Green Monkey: Chlorocebus_sabeus 1.1/chlSab2 (https://hgdownload.soe.ucsc.edu/goldenPath/chlSab2/bigZips/chlSab2.fa.gz). Gene annotations were downloaded from GENCODE (Human v34, Mouse v25, https://www.gencodegenes.org/), and disease-risk variants from the GWAS Catalog mapping to hg38 were download from the UCSC Genome Browser (https://hgdownload.soe.ucsc.edu/goldenPath/hg38/database/gwasCatalog.txt.gz).
Liftover files for mapping between mouse strains were download from http://hgdownload.cse.ucsc.edu/goldenpath/mm10/liftOver/mm10ToGCA_001624865.1_SPRET_EiJ_v1.over.chain.gz (C57Bl/6/mm10 to SPRET) and https://hgdownload.soe.ucsc.edu/goldenPath/GCA_001624865.1_SPRET_EiJ_v1/liftOver/GCA_001624865.1_SPRET_EiJ_v1ToMm10.over.chain.gz (SPRET to C57Bl/6/mm10). Per-chromosome VCF files containing genotyping data for the samples analyzed in Kristjánsdóttir et al 63 were downloaded from the 1000 Genomes Project (ftp://ftp.1000genomes.ebi.ac.uk/vol1/ftp/data_collections/1000G_2504_high_coverage/working/20201028_3202_raw_GT_with_annot/).

# Field-specific reporting

Please select the one below that is the best fit for your research. If you are not sure, read the appropriate sections before making your selection.

☒ Life sciences ☐ Behavioural & social sciences ☐ Ecological, evolutionary & environmental sciences

For a reference copy of the document with all sections, see nature.com/documents/nr-reporting-summary-flat.pdf

# Life sciences study design

All studies must disclose on these points even when the disclosure is negative.

| Sample size | No statistical methods were used to predetermine sample size. Nearly all assays performed in this study were profiled as biological replicates (n=2), starting from distinct biological material grown and treated as distinct samples. Resulting sequencing data was normalized and analyzed using R/DESeq2, and if more than 1% of features were considered differentially regulated using an FDR cutoff of 5% (or otherwise stated in the manuscript) we determined the experiment had a sufficient number of replicates. Please see Table S1 for an overview of samples collected and their replicate annotation. |
|---|---|
| Data exclusions | No data was excluded in the context of this study. |

| Replication | A minimum of duplicate experiments was performed for all experiments, and all attempts at replication were successful. In some experiments, additional replicates were part of the experimental design, such as the replicate inserts with differing barcodes in the TSS-MPRA experiments, which also replicated across the duplicates. For the ChIP-seq experiments, technical replicate experiments were performed with different antibodies on the same batch of cells. Replicability of the peak read count distributions across the genome between replicates, as well as the similarity of motif enrichment scores derived from peaks above the significance threshold that were obtained when using different antibodies on the same cells indicated a high level of replicability of the genome-wide transcription factor location analyses |
| --- | --- |
| Randomization | Randomization was not relevant to our study, and experiments were not randomized. |
| Blinding | No interventions that would require blinding to exclude bias were performed.<br>The investigators were not blinded to allocation during experiments and outcome assessment. |

# Reporting for specific materials, systems and methods

We require information from authors about some types of materials, experimental systems and methods used in many studies. Here, indicate whether each material, system or method listed is relevant to your study. If you are not sure if a list item applies to your research, read the appropriate section before selecting a response.

## Materials & experimental systems

| n/a | Involved in the study |
| --- | --- |
| ☐ | ☒ Antibodies |
| ☐ | ☒ Eukaryotic cell lines |
| ☒ | ☐ Palaeontology and archaeology |
| ☒ | ☐ Animals and other organisms |
| ☒ | ☐ Human research participants |
| ☒ | ☐ Clinical data |
| ☒ | ☐ Dual use research of concern |

## Methods

| n/a | Involved in the study |
| --- | --- |
| ☐ | ☒ ChIP-seq |
| ☒ | ☐ Flow cytometry |
| ☒ | ☐ MRI-based neuroimaging |

## Antibodies

| Antibodies used | anti YY1 (western, Santa Cruz, sc-7341 HRP  YY1 (H-10)), anti YY1 (ChIP, ActiveMotif AB_2793763), anti NRF1 (Abcam , ab55744), anti β-Actin (Cell Signaling D6A8 - 8457S, Rabbit mAb #8457), and anti HA (Abcam, ab9110)<br><br>Antibody dilution for western blots:<br>anti-NRF1 (ab55744), 1:1000<br>anti-YY1 (sc-7341) 1:200<br>anti-beta actin (Cell Signaling D6A8 - 8457S, Rabbit mAb #8457) 1:2500<br><br>For ChIP-seq:<br>anti-HA (Abcam ab9110), rabbit pAb, 2 µg for 1x106 cells<br>anti-NRF1 (ab175932), rabbit mAb,  2 µg for 2.5x106 cells<br>anti-YY1 (Active Motif 61980, RRID: AB_2793763), rabbit pAb,  2 µg for 2.5x106 cells |
| --- | --- |
| Validation | Commercial validated monoclonal and polyclonal antibodies were further assessed by apparent molecular weight in western blot of the detected proteins across multiple cell lines (Supplementary Fig. 4a). Anti-NRF1 (ab55744) recognizes human NRF1 aa 201-285. anti-YY (sc-7341) was raised against full-length human YY1. The rabbit antibodies used for ChIP-seq were validated by ChIP-seq with additional mouse monoclonal antibodies that recognize the same antigens (anti-HA (Biolegend 901501), anti-NRF1 (Diagenode C15200013), anti-YY1 (Diagenode C15410345)), which resulted in near-identical ChIP-seq enrichment patterns. |

## Eukaryotic cell lines

Policy information about cell lines

| Cell line source(s) | Cell lines used in this study were gifted from Aaron Carlin's lab (U2OS, Vero E6), Xiangdong Fu's lab (HepG2), and James Kadonaga's Lab (HEK293T) from University of California, San Diego. |
| --- | --- |
| Authentication | Cells lines were verified by the presence of cell-line specific DNA variants and by the phenotypic similarity of their genomics data to published resources (e.g. TSS locations identified in e.g. U2OS cells match open chromatin regions in U2OS cells identified by the ENCODE consortium). |
| Mycoplasma contamination | Cells were routinely tested for mycoplasma contamination. All tests were negative. |
| Commonly misidentified lines<br>(See ICLAC register) | No commonly misidentified cell lines were used in the study. |

# ChIP-seq

## Data deposition

☒ Confirm that both raw and final processed data have been deposited in a public database such as GEO.

☒ Confirm that you have deposited or provided access to graph files (e.g. BED files) for the called peaks.

| | |
|---|---|
| Data access links<br>*May remain private before publication.* | All raw and processed data generated for this study can be accessed at NCBI Gene Expression Omnibus (GEO; https://www.ncbi.nlm.nih.gov/geo/) accession number GSE199431. |
| Files in database submission | ChIP-seq peak file provided as a supplemental file: GSE199431_peaks.U2OS.dnNRF1.HA.bed.gz, GSE199431_peaks.U2OS.NRF1.bed.gz, GSE199431_peaks.U2OS.YY1.bed.gz |
| Genome browser session<br>(e.g. UCSC) | ChIP and csRNA-seq data are included for genomes hg38 and mm10:<br>https://genome.ucsc.edu/s/Cbenner/Reviewer%2DSession%2D240306%2DTfMotifGrammar |

## Methodology

| | |
|---|---|
| Replicates | Only a single experiment was performed. |
| Sequencing depth | >10 million reads |
| Antibodies | anti YY1 (ChIP, ActiveMotif AB_2793763), anti NRF1 (Abcam , ab55744), and anti HA (Abcam, ab9110), which was used to target a dominant negative NRF1 protein tagged with HA. |
| Peak calling parameters | Peaks were found using HOMER's findPeaks program: findPeaks IP_Data/ -i Input_Data -style factor -o auto |
| Data quality | 6,548 (NRF1), 9,454 (YY1) and 28,450 (dnNRF1) total peaks were identified, and the top de novo motif identifed using HOMER's motif analysis program for each experiment was a match for either the NRF1 or YY1 motif. |
| Software | HOMER v5.0 was used for the ChIP-seq analysis. |

