## [Peer Review File · Nature]

Manuscript Title: Position-dependent function of human sequence-specific transcription factors

Reviewer Comments & Author Rebuttals

Reviewer Reports on the Initial Version:

Referees' comments:

Referee #1 (Remarks to the Author):

The manuscript by Duttke and colleagues investigates the role that the position of transcription factor (TF) binding relative to the transcription start site (TSS) plays in gene regulation. The authors note a strong positional dependence between TF binding sites and the TSS. Using a series of experiments, the authors argue that the TF binding location relative to the location of the TSS has a strong effect on the activity of the transcription factor, including whether it will act as a transcriptional activator or repressor.

Major issues:

-- The idea that transcription factors exhibit a strong positional dependence is not (on its own) all that novel. There are a number of papers that the authors should have cited in their work as precedence for this finding, including papers by the Lander (Grossman et al., PNAS, 2018) and Lis labs (Core et al., Nat. Genet., 2014; Tippens et al., Nat. Genet., 2020). There are likely other reports as well. Notably, Core et al. (Nat. Genet., 2014) notes the positional differences between transcriptional activators and repressors also discovered here. This does not mean there is no room for the present study -- just that existing work should be cited, rather than ignored.

-- The heatmaps used to argue that NRF1 motifs are enriched downstream of the TSS for TSSs that increased following NRF1 knockdown look like there are many more down-regulated than up-regulated genes. But in the text the number of genes looks symmetrical between up/ down regulation. How do the authors' reconcile this apparent disparity?

-- The argument that NRF1 binding motifs are enriched downstream of genes that are up-regulated following siNRF1 is mainly supported by heatmaps. However, it looks like this effect may be driven by just a few of the rows in the heatmap. This is not surprising on its own: knockdown experiments will have large numbers of secondary effects since they measure changes after days of knockdown (incidentally, I encourage the authors to modify the text arguing that having up-regulated genes is a surprise on line 90; this has been observed for many transcriptional activators, even for direct effects from faster perturbations). However, the authors need to complement heatmaps with a more quantitative analysis of enrichment that would not be fooled by just a few genes that have a large number of NRF1 binding sites downstream of the TSS.

-- Are the candidate repressive NRF1 motifs downstream of the TSS actually bound by NRF1? This should be shown using existing or novel ChIP-seq data.

-- Currently there is virtually no quantitative analysis of the signal shapes in the MPRA experiments. Assay noise needs to be rigorously assessed in the MPRA experiments before any conclusions can be drawn.

Referee #2 (Remarks to the Author):

This manuscript investigates to what extent the position of a TF binding site relative to the TSS modulates its effect on transcriptional initiation. To this end, the authors use an experimental assay named TSS-MPRA which allows them to investigate this question from a number of complementary angles, such as (i) endogenous transcriptional activity, (ii) impact of noncoding polymorphisms that disrupt TF binding sites, (iii) activity of promoter constructs with planted TF binding sites. A main conclusion is that each TF has a unique spatial profile as far as its effect on expression is concerned, and that this behavior is consistent across these angles. Even though the approach is simple, and combines existing techniques, this comprehensive analysis should be interesting for many researchers working in regulatory genomics.

Major issues:

At a technical level, I have one major concern: First, the spatial profile in terms of simple A/C/G/T base composition relative to the TSS could have biases, but nowhere in the paper is this simple analysis performed. Furthermore, transcriptional activity could be correlated with this simple base composition, in a way that potentially depends on position. In other words, before analyzing the association between TF motif occurrence and TSS-MPRA signal, the authors should first have the effect of base composition on transcription, as it could be a huge confounding factor: Each TF motif has its own base composition biases, and therefore a TF motif can act as a proxy for base composition. It is not unthinkable that the structure in Figure 2f is simply a reflection of this motif bias and proxy effect. This needs to be ruled out rigorously.

Note that the fact that an observed/expected ratio is shown for the enrichment profiles in Figure 1b may or not give a false sense of reassurance, depending on what exactly was done. The phrase "normalizing these patterns by the motif's background frequency" on line 66 on page is not precise enough: If a single average motif density was used to normalize, this still leaves the possibility that the enrichment patterns are an indirect reflection of spatial structure in the base composition distribution relative to the TSS. The normalization would have to be done in a position-specific way.

Beyond this basic confounding effect, care must also be taken with the mechanistic interpretation of any identified descriptive correlations. For instance, nucleosome occupancy is known to strongly depend on low-complexity DNA sequence features such as poly-A/poly-T stretches, CG-stretches, etc, which again can be confounded with the specific base preference of a motif for any given TF. This nuance does not seem to be reflected anywhere in the manuscript.

Other comments:

- In general, the use of the word “motif” is sloppy in this ms: Most experts would use this for the IUPAC consensus or PWM that captures the general sequence preference of a given DNA binding domain. An instance of a match to such a “motif” is not a “motif” but a “motif match” / “TF binding site” / “cis-regulatory element”.
- line 21: best to avoid “novel” here since TSS-MPRA is a minor variation on existing assays, as the authors properly acknowledge in line 182
- line 22: “here we” -> “we here”
- line 22: “outcome” -> “effect”
- line 24: there should not be a comma between “preferential” and “helical”
- line 25: “canonical” -> “nominal” ?
- line 28: “TF motifs” -> “TF binding site” (one of several instances; see above)
- line 49: why use a new term “TSR” when we already have “cis-regulatory module (CRM)” and when it seems misleading to refer to enhancers by what is maybe just a by-product of their key function (namely, transcription of eRNA).
- lines 89-90, 152-153 : in each instance, need to provide a p-value and test used to back up this claim
- line 151: provide citation for HOMER
- line 155: remove “and” at end of sentence
- line 163: replace “transcription repression” with “reduction in transcription”
- line 231: “based on the consensus of their unique properties” : no idea what this means...
- lines 253-272: It may be nice to connect some of this discussion to some of the older literature on models for predicting gene expression from sequence that were developed in the days of DNA microarrays before next-generation sequencing (e.g., PMID 15084257, 12626739, 11175784, 28679386, and many papers building on those approaches)
- Code Availability section: provide versions of software, and references if available
- line 587: remove “also”
- line 593: no comma after “individual”
- line 687: not clear what “+1” in this IUPAC consensus denotes; also, best to explicitly say this is

IUPAC motif, or define degenerate base symbols explicitly

- line 690: please provide version / is this a co-author of present study? If not, please provide reference.

- Note that when printed to paper from Preview on Mac, all figures are severely corrupted and print as a square black box. Printing from Acrobat is okay, and figure look fine on screen.

- Figure 3c: Is the y-axis perhaps a log-ratio instead of a "ratio"?

Referee #3 (Remarks to the Author):

This is an interesting study examining the effect of transcription factor binding site (TFBS) position on transcription. The major conclusion of this study is that, for some TFs, the location of their TFBS relative to the transcription start site (TSS) has a profound effect on transcript levels (e.g., activating vs repressing). To draw this conclusion, the authors take advantage of the relatively new csRNA-seq method they developed, as well as a completely new reporter assay technique they call TSS-MPRA. The combination of these two assays with careful bioinformatics analysis offers the most detailed insight to date into TFBS function with respect to relative TSS position. For example, Fig. 2f provides, to my knowledge, the first ever look at how the position of TFBS for the majority of TFs impacts the effect of their binding on transcription. Upon reviewing this work, I have one conceptual issue with this paper, several issues with data interpretation/presentation, and some technical issues.

Major comments

1. (Conceptual) The basic premise that TFBS location relative to the TSS can impact gene expression has been around for a long time. For example, in bacterial systems we have known for several decades that repressors often bind downstream of the TSS to block transcription, while activators are normally just upstream of the TSS. Likewise, the YY1 vertebrate TF highlighted in this study is well known to both activate and repress transcription (this was the origin of its name – Ying Yang). The power of this current study is that we really get to see this effect on a systematic, global scale, for a range of vertebrate TFs using a range of experimental and computational approaches. So, the contribution of this study to the scientific literature is still important, but it is somewhat dampened because its major conclusion does not lead to brand new concepts in gene regulation.

2. Line 56, "Exploiting these TSSs as anchors for sequence analysis revealed that the motifs of many but not all TFs naturally displayed preferential positioning relative to sites of transcription initiation": This is a potential over-generalization, and we can't tell from the provided data exactly what this statement means. The authors need to clearly state in the results how many motifs for how many TFs were actually tested here (Figures and SFigures both only show possibly cherry picked examples). Otherwise, it is impossible to gauge exactly what is meant by "many but not all"; this is a qualitative statement that could at least be semi-quantitative (i.e., set a reasonable cutoff, and tell us that XX out of YY tested TFs display this behavior). Also, were these TFs chosen based on prior knowledge, or were many TFs tested in a completely unbiased fashion, and then only these few

examples are shown? These important details need to be clearly indicated in the results section.

3. Another over-generalization, “Motif distribution patterns were more pronounced in the absence of core promoter elements (Fig. 1c, Fig. S2b).”: The statement makes it sound like it applies to all motifs, but evidence is only shown for a single TF (NFY).

4. Line 58, “Most patterns included a 10-bp periodicity”: where is the evidence for this? Need some sort of calculation to classify which motifs are helical and which are not. There must be some simple way to gauge how close the observed periodicity is to the expected 10-11 base periodicity of DNA. The authors need to be much more systematic and quantitative to support this claim. Currently there is no evidence that I can find, even on a qualitative level.

5. Fig. S3b is good to see, but it is hard to fully understand the y-axis. Would these regions actually be called ChIP-seq “peaks” or are we squinting at noise? To complement the current analysis I would like to see a simple enrichment analysis where these regions are binned and the over-representation of NRF1 ChIP-seq peaks is estimated within these bins. Same thing for Fig. S5c.

6. Based on my understanding of how csRNA-seq works, the signal should be capturing multiple types of RNAs, including enhancer RNAs, pri-miRNAs, antisense transcripts, and promoter transcripts. Is there a way to estimate the relative proportion of these four “flavors” of RNAs in these analyses? TFBS location might have a different effect on eRNAs vs promoter RNAs, for example. Likewise, if eRNAs are underrepresented or have weaker/noisier signal, for example, the observed TFBS patterns could be dominated by promoters.

7. Fig. 2f is a very rich figure with lots of useful information. The underlying data need to be provided in the supplement for additional data mining purposes. For example, I would love to know what that class of TFs is that represses transcription specifically at a distance of around 50-bp upstream of the TSS.

8. Line 267, “Position-dependent interactions were observed among many motifs including CRE, Ebox, ETS, GFX, GFY, NRF1, NFY, Sp1 and YY1 but not all (i.e. LRF, FigS. 15b).”: Where are the data supporting this claim? For example, I don’t see any mention of E-boxes in Fig. 4 or Fig. S15.

9. Little of no QC is provided for the functional genomics data – did the ChIP work? Did RNA-seq? Are there good QC metrics for csRNA-seq? ENCODE-type standards need to be applied and provided. For example, for ChIP-seq, I would need to see #unique mapped reads, peak counts, FRiP scores, and motif enrichment for the ChIP’d TF (relative to e.g. all motifs in the HOMER library) at minimum.

10. Application of the TSS and TF motif analysis to genetic polymorphisms in humans would increase the impact of this manuscript and add a disease genetic risk component. While a comprehensive application might be outside of the scope of this manuscript, consider applying this approach to a single disease’s genetic risk variants in the context of a disease appropriate cellular context for which the lab already has csRNA-seq.

Minor comments

1. Please provide the quantified level of TF knockdown for each siRNA in Table S1.
2. Refs 1 and 2 do not appear to support the statement they are meant to support (one is on flower development in plants and the other involves fly segment development). These are likely mistakes. Ref 3 makes sense at least.
3. Line 40, typo: 'TF motifs pacing'.
4. Fig. S1c is never really described in the text. I can't tell what it is showing. What exactly is MEIRLOP doing?
5. Fig. S1f is also never described in the text, but the results look interesting and important.
6. Official gene names should be included where possible (for example, LRF=ZBTB7A).
7. Line 104: I think the figure refs should be Fig. 1b,d (and not c,d).
8. Calling YY1 an activator is misleading; its name (Ying Yang 1) is based on the fact that it can activate or repress transcription.
9. Fig. 2: It would be a lot less confusing if the red "activates" label was on top and the blue "represses" label was on the bottom (currently they are in the opposite order – the colors match up but not their positions).
10. Line 154: Should be (Fig. 2f).
11. Citation on line 182 is messed up.
12. Fig. 3c might be clearer if the y-axes had identical ranges for all of the panels.
13. The x-axis label in Fig. 4a is misleading; I think it should range from -100 to +40. Same with S12 all panels.
14. Fig. S7 is very hard to read.
15. The FAQ for TSS-MPRA is potentially useful. Suggestions below:
 - a. The FAQ currently reads more as a very helpful blog than manuscript methods protocol. In multiple sections, the FAQ shares the success or lack of success for specific approaches – this is likely to change over time. I would suggest instead posting this online and providing a url in the manuscript to provide an opportunity to update the protocols/FAQ as additional considerations and optimizations are available.
 - b. If this will be part of the manuscript, more details are needed. For example, "ensure you have a good transfection rate" needs to include the transfection rate that would be considered "good".

Under “how much RNA can we use as input?” be specific about the higher ratio increasing or decreasing sensitivity – rather than using the non-directional “influence”.

c. Under “how diverse can my plasmid library be?”, consider defining diversity and complexity. Instead of presenting a cost-based argument, consider simply providing the amount of sequencing needed and allowing readers to make their own cost-based assessments.

d. It was unclear what “nascent style” referred to in the section “Can TSS-MPRAs be performed nascent style?”.

e. Many abbreviations are included without definition. While most of these are common experimental phrases, definitions are needed for clarity.

Author Rebuttals to Initial Comments:

Referees' comments:

Referee #1 (Remarks to the Author)

The manuscript by Duttke and colleagues investigates the role that the position of transcription factor (TF) binding relative to the transcription start site (TSS) plays in gene regulation. The authors note a strong positional dependence between TF binding sites and the TSS. Using a series of experiments, the authors argue that the TF binding location relative to the location of the TSS has a strong effect on the activity of the transcription factor, including whether it will act as a transcriptional activator or repressor.

Thank you for this precise and accurate summary.

Major issues:

The idea that transcription factors exhibit a strong positional dependence is not (on its own) all that novel. There are a number of papers that the authors should have cited in their work as precedence for this finding, including papers by the Lander (Grossman et al., PNAS, 2018) and Lis labs (Core et al., Nat. Genet., 2014; Tippens et al., Nat. Genet., 2020). There are likely other reports as well. Notably, Core et al. (Nat. Genet., 2014) notes the positional differences between transcriptional activators and repressors also discovered here. This does not mean there is no room for the present study -- just that existing work should be cited, rather than ignored.

We fully agree with the reviewer that these three papers as well as others [perhaps most notably Avsec et al 2021 Nat.Gen] report that different TFs can have distinct positional binding preferences. We have augmented the manuscript to point this out and clarified the advance we contributed, which is that we revealed that the function of the *same* transcription factor can change depending on its position. The above mentioned, and to the best of our knowledge all other manuscripts, describe positional binding preferences of *different* transcription factors. Furthermore, our study is unique in that it provides multiple lines of functional evidence to support these claims. Aside from now citing these papers to give proper visibility, we also refined our language to emphasize that we identified *why and how* position has an impact and thus, revealed predictable rules for this phenomenon.

[detailed response:]

Core et al. 2014 classifies ALL transcription factors into either “central-binding factors (for example, SP1) and TSS-proximal binding factors (for example, PML)” while Grossman et al 2019 distinguishes six distinct profiles based on binding preference profiles in open chromatin. Tippens et al. 2020 reports (Fig.3) “Interestingly, some motifs were well aligned to TSS, especially those known to recruit and position TFIID. Similar to the well-known TATA box bound by TBP (maximum motif density at -32 bp), SP1 (ref. 24) (at -53 bp) and STAT2 (ref. 32) (-5 bp) show striking TSS alignment and are known to recruit TFIID.” None of these papers report evidence that position is critical or determines the function of that individual transcription factor.

Related to activators or repressors, Core et al. 2014 reports “Most factors bind between the two divergent TSSs (central binders), suggesting that they have a role in activation and are likely a major determinant or result of the overall architecture of initiation sites. In contrast, TSS-proximal transcription factors are primarily enriched for repressors, suggesting that certain repressors can act by preventing access of the transcription machinery to critical parts of the core promoter.” By contrast, we argue that the function of the *same* transcription factor is dependent on the position. For example YY1 binding to the proximal region in Core et. al. is activating while binding in this central region is repressive. Even TFs that are exclusively activating or repressing show evidence that their specific location has an impact on function and indeed most repressors are more potent when binding the central region rather than the proximal regions. In other words, we show that the TF’s relative potency or its ability to activate

or repress depends on its binding position. Aside from notable differences in the resolution and identified areas, the above mentioned papers argue that different TFs or TF families have binding preferences and are repressors OR activators.

We thank the reviewer for pointing this out and hope the revised manuscript better highlights the novelty of our findings while also giving proper credit to prior advances.

The heatmaps used to argue that NRF1 motifs are enriched downstream of the TSS for TSSs that increased following NRF1 knockdown look like there are many more down-regulated than up-regulated genes. But in the text the number of genes looks symmetrical between up/ down regulation. How do the authors' reconcile this apparent disparity?

We apologize for the inconsistency. Differential regulation was based on the DESeq2 results while the red and green triangles in Fig. 1d,f (Now Fig. 1e,g) were added by us with the goal to improve readability of the figure (MEPP-plot). We adjusted the figure to be consistent with the DESeq2 results.

The argument that NRF1 binding motifs are enriched downstream of genes that are up-regulated following siNRF1 is mainly supported by heatmaps. However, it looks like this effect may be driven by just a few of the rows in the heatmap. This is not surprising on its own: knockdown experiments will have large numbers of secondary effects since they measure changes after days of knockdown (incidentally, I encourage the authors to modify the text arguing that having up-regulated genes is a surprise on line 90; this has been observed for many transcriptional activators, even for direct effects from faster perturbations). However, the authors need to complement heatmaps with a more quantitative analysis of enrichment that would not be fooled by just a few genes that have a large number of NRF1 binding sites downstream of the TSS.

We have adjusted the language in line 90 (now line 98) and acknowledge that TSS up-regulated by NRF1 knockdown are likely a mix of sites regulated by secondary effects and those directly impacted by downstream NRF1 binding sites. However, multiple lines of evidence support a direct role for the downstream NRF1 sites in TSS regulation: 1) Positional correlation analysis by MEPP supports an association between NRF1 binding sites and up-regulated TSSs in the downstream regions that is well outside the 95% confidence intervals (Fig. 1e, lower panel, dotted lines), 2) we now present ChIP-seq data for NRF1 in U2OS cells showing that the ChIP-seq peak summits that describe the locations of NRF1 binding are enriched in the downstream region of these regulated TSSs (Fig. 1f, Fig. S6c), and 3) these TSS are not up-regulated in the presence of dominant negative NRF1 (Fig. 1g, S9c,d), which makes them less likely to be regulated in trans following a loss of active NRF1.

Are the candidate repressive NRF1 motifs downstream of the TSS actually bound by NRF1? This should be shown using existing or novel ChIP-seq data.

We thank reviewer 1 for this suggestion. As mentioned above, we have generated additional new ChIP-seq data for NRF1 and YY1. When stratifying the TSSs based on whether they are up- or downregulated by TF knockdown, the ChIP-seq peak summit locations for these factors are located down- and upstream of the TSS, similar to their motif positions (Fig 1f, S8b). Together, this demonstrates that these factors indeed bind their motifs in a function-specific fashion.

Currently there is virtually no quantitative analysis of the signal shapes in the MPRA experiments. Assay noise needs to be rigorously assessed in the MPRA experiments before any conclusions can be drawn.

We thank the reviewer for pointing this out and we apologize for the missing information. Thorough QC analysis was performed but we had only included the previous Fig S8, which shows reproducibility in TSS level and previous Fig S9, which shows the impact of using diverse barcodes. We have now further expanded our assessment of the variation in our TSS-MPRA data (now Fig. S13, S14, S16, S17, S23). TSS-MPRA inserts were designed with at least 4 independent DNA barcodes to make sure the barcode sequences did not influence our conclusion (Fig. S14, S17, S23). First, we assessed how reproducible the RNA counts from each insert-barcode pair were across biological replicates (Fig. S14a,d, 23a). Second, we compared how reproducible each insert was across each of the different barcodes (Fig. S14b,c,e,f, S23b,c). Finally, to provide evidence that our conclusions were not impacted by outliers, we reproduced the assessment of TSS levels using each set of barcodes individually, finding nearly identical results, suggesting the TSS-MPRA results are robust and reproducible (e.g. Fig. S17).

Referee #2 (Remarks to the Author):

This manuscript investigates to what extent the position of a TF binding site relative to the TSS modulates its effect on transcriptional initiation. To this end, the authors use an experimental assay named TSS-MPRA which allows them to investigate this question from a number of complementary angles, such as (i) endogenous transcriptional activity, (ii) impact of noncoding polymorphisms that disrupt TF binding sites, (iii) activity of promoter constructs with planted TF binding sites. A main conclusion is that each TF has a unique spatial profile as far as its effect on expression is concerned, and that this behavior is consistent across these angles. Even though the approach is simple, and combines existing techniques, this comprehensive analysis should be interesting for many researchers working in regulatory genomics.

We thank Reviewer 2 for these positive comments!

Major issues:

At a technical level, I have one major concern: First, the spatial profile in terms of simple A/C/G/T base composition relative to the TSS could have biases, but nowhere in the paper is this simple analysis performed. Furthermore, transcriptional activity could be correlated with this simple base composition, in a way that potentially depends on position. In other words, before analyzing the association between TF motif occurrence and TSS-MPRA signal, the authors should first have the effect of base composition on transcription, as it could be a huge confounding factor: Each TF motif has its own base composition biases, and therefore a TF motif can act as a proxy for base composition. It is not unthinkable that the structure in Figure 2f is simply a reflection of this motif bias and proxy effect. This needs to be ruled out rigorously.

We thank Reviewer 2 for bringing up this very important point. It is difficult to distinguish the relative impact of positional bias in nucleotide content from TF binding sites (as now shown in Fig 1b). Furthermore, it is clear from the analysis of genetic variants that specific nucleotides are associated with increased TSS activity in a position dependent manner (now shown in Fig. S10). Both of these biases suggest an important role for core promoter elements in dictating TSS strength given the strong biases present near -30 bp (i.e. TATA), +1 (i.e. Initiator), and the region downstream of the TSS (i.e. DPE, DPR). To address this point we developed HOMER2, which attempts to score TF motif enrichment independent of positional nucleotide bias.

HOMER2 includes two new analysis strategies: First, when scoring motif enrichment, HOMER2 selects a set of empirical background sequences from the genome that matches the positional nucleotide bias present at TSS (Fig. S1,S2). This allows us to more accurately assess the relative enrichment or depletion of TF binding sites relative to the TSS (e.g. Fig. 1d). Second, when assessing genetic variants, HOMER2 models the impact that

positional distribution of transcription-altering variants (i.e. Fig. S10) has on expected TF binding site scores, improving the rigor of our calculations associating genetic variants with position-dependent TF function (Fig. 2b-f, etc.). We also note that these tools are general in nature and can be applied to other studies where positional analysis of sequence features is critical.

One change we want to note is that the new analysis of natural genetic variation between mouse strains is now limited to single nucleotide variants to enable the correction of position-dependent variant bias (Fig. 2f, Fig. S10). The original analysis strategy also incorporated indels and structural variation, increasing the sensitivity of the analysis, but unfortunately lacking more rigorous controls. Both are now included in Fig. S12 for comparison.

Note that the fact that an observed/expected ratio is shown for the enrichment profiles in Figure 1b may or not give a false sense of reassurance, depending on what exactly was done. The phrase “normalizing these patterns by the motif’s background frequency” on line 66 on page is not precise enough: If a single average motif density was used to normalize, this still leaves the possibility that the enrichment patterns are an indirect reflection of spatial structure in the base composition distribution relative to the TSS. The normalization would have to be done in a position-specific way.

Beyond this basic confounding effect, care must also be taken with the mechanistic interpretation of any identified descriptive correlations. For instance, nucleosome occupancy is known to strongly depend on low-complexity DNA sequence features such as poly-A/poly-T stretches, CG-stretches, etc, which again can be confounded with the specific base preference of a motif for any given TF. This nuance does not seem to be reflected anywhere in the manuscript.

As described in our response to the previous comment, we have developed HOMER2 to help us better model the positional sequence content near the TSS enabling us to more accurately calculate the expected frequency of TF binding sites at a given position. We now assess the positional enrichment or depletion of motifs (e.g. Fig 1c,d) by comparing their occurrences at the TSS to genomic background sequences that match the positional sequence content and overall GC% distribution of TSS (Fig. S1-2).

While HOMER2 can select backgrounds that account for lower-order sequence contexts (i.e. we control for dinucleotide frequencies [k=2] in the manuscript), higher order features (e.g. longer poly-A/T stretches) and their potential cross identification/shared properties with TF motifs are more difficult to account for. This is one of the reasons we strived to investigate TF binding site positioning using a variety of different assays (including functional perturbation and ChIP-seq), data sources, and analytical techniques.

Other comments:

In general, the use of the word “motif” is sloppy in this ms: Most experts would use this for the IUPAC consensus or PWM that captures the general sequence preference of a given DNA binding domain. An instance of a match to such a “motif” is not a “motif” but a “motif match” / “TF binding site” / “cis-regulatory element”.

Thank you for pointing this out. We have revised the manuscript accordingly.

line 21: best to avoid “novel” here since TSS-MPRA is a minor variation on existing assays, as the authors properly acknowledge in line 182

Done

line 22: “here we” -> “we here”

Done

line 22: “outcome” -> “effect”

Done

line 24: there should not be a comma between “preferential” and “helical”

Done, thank you for catching this!

line 25: “canonical” -> “nominal” ?

We use canonical here in the sense of canonical vs. non-canonical elements. As such, we believe nominal is not the right choice but are welcoming alternative suggestions.

line 28: “TF motifs” -> “TF binding site” (one of several instances; see above)

Done, thank you for pointing this out.

line 49: why use a new term “TSR” when we already have “cis-regulatory module (CRM)” and when it seems misleading to refer to enhancers by what is maybe just a by-product of their key function (namely, transcription of eRNA).

We welcome this comment and revised the manuscript to more accurately rationalize our use of TSR while at the same time minimize its use where we think it might cause confusion. Of note, “TSR” is not a new term or coined by us (e.g., PMID: 35947745, PMID: 32597978 etc.). The difference between TSRs and *cis*-regulatory elements (CREs) is that CREs modulate the expression of genes and may or may not be transcribed. By contrast, we define “transcription start regions” in this manuscript based on direct evidence of transcription initiation. While most TSRs in promoter-distal regions have all of the *bona fide* hallmarks of enhancers, we do not have functional evidence that they do and which gene they modulate. Thus, while some scientists would call these regions CREs and yet others enhancers, we agree with what is advocated by Halfon 2019 (PMID: 30553552) and others that it is important to distinguish regions defined by genomic data from those defined by functional assays such as CRISPR or reporter/GUS-staining. The revision is as follows:

“Regulatory elements including promoters and enhancers as defined by active transcription initiation, that we hereafter collectively refer to as transcription start regions (TSRs, Halfon 2019), often start transcription from several different TSS locations rather than a single site.”

lines 89-90, 152-153 : in each instance, need to provide a p-value and test used to back up this claim

We have now updated these results to reflect the approach and provide the p-value.

line 151: provide citation for HOMER

Done

line 155: remove “and” at end of sentence

Done, thank you for pointing this out.

line 163: replace “transcription repression” with “reduction in transcription”

Done

line 231: “based on the consensus of their unique properties” : no idea what this means...

We thank the reviewer for pointing this out and we apologize if this sentence was not clear. We tried to communicate that the distinct spatial-functional profiles of different TFs jointly guide TSS selection. We edited

the sentence to “*These findings propose a model whereby human TFs can directly guide transcription initiation based on the consensus of their unique spatial-functional profiles (Fig. S18e).*”

lines 253-272: It may be nice to connect some of this discussion to some of the older literature on models for predicting gene expression from sequence that were developed in the days of DNA microarrays before next-generation sequencing (e.g., PMID 15084257, 12626739, 11175784, 28679386, and many papers building on those approaches)

We appreciate pointing out these classic papers. We did not cite or connect to them as 1) the success of predicting gene expression in yeast (PMID 15084257, 12626739, 11175784) has been quite different from what has been achieved in humans, 2) as these papers do not discuss TSS or transcription initiation and 3) we want to keep this manuscript focused on the regulation of mammalian transcription initiation. As such, and in light of the journal's tight space limitations, we only briefly touch on efforts related to predicting gene expression from DNA in humans in line 141 and 333 for which we think PMID: 27197224 and PMID: 15131651 provide more appropriate references.

Code Availability section: provide versions of software, and references if available

Thank you for this suggestion. We have now added the used versions of the software.

line 587: remove “also”

Done

line 593: no comma after “individual”

Done, thank you for catching this!

line 687: not clear what “+1” in this IUPAC consensus denotes; also, best to explicitly say this is IUPAC motif, or define degenerate base symbols explicitly

The sentence was edited. Changes are shown in *italics* “To segregate TSS based on the presence of Initiator core promoter elements, the genomic sequence adjacent to each TSS was scanned for a strand and position-specific match to the sequence *IUPAC motif* BBCA+1BW (*where the A+1 defines is on the initiating nucleotide*) “

line 690: please provide version / is this a co-author of present study? If not, please provide reference.

Our apologies if we were not clear here. Yes, MEPP was developed by co-author Nathaniel Delos Santos. It is now released [PMID: 36267125]. The github link is provided in the methods.

Note that when printed to paper from Preview on Mac, all figures are severely corrupted and print as a square black box. Printing from Acrobat is okay, and figure look fine on screen.

We are sorry to hear. We tested the latest pdf and hope no issues will arise.

Figure 3c: Is the y-axis perhaps a log-ratio instead of a “ratio”?

Yes, log₂ ratio. We thank the reviewer for pointing this out.

Referee #3 (Remarks to the Author):

This is an interesting study examining the effect of transcription factor binding site (TFBS) position on transcription. The major conclusion of this study is that, for some TFs, the location of their TFBS relative to the transcription start site (TSS) has a profound effect on transcript levels (e.g., activating vs repressing). To draw this conclusion, the authors take advantage of the relatively new csRNA-seq method they developed, as well

as a completely new reporter assay technique they call TSS-MPRA. The combination of these two assays with careful bioinformatics analysis offers the most detailed insight to date into TFBS function with respect to relative TSS position. For example, Fig. 2f provides, to my knowledge, the first ever look at how the position of TFBS for the majority of TFs impacts the effect of their binding on transcription. Upon reviewing this work, I have one conceptual issue with this paper, several issues with data interpretation/presentation, and some technical issues.

Thank you for these positive and constructive comments!

Major comments

1. (Conceptual) *The basic premise that TFBS location relative to the TSS can impact gene expression has been around for a long time. For example, in bacterial systems we have known for several decades that repressors often bind downstream of the TSS to block transcription, while activators are normally just upstream of the TSS. Likewise, the YY1 vertebrate TF highlighted in this study is well known to both activate and repress transcription (this was the origin of its name – Ying Yang). The power of this current study is that we really get to see this effect on a systematic, global scale, for a range of vertebrate TFs using a range of experimental and computational approaches. So, the contribution of this study to the scientific literature is still important, but it is somewhat dampened because its major conclusion does not lead to brand new concepts in gene regulation.*

Thank you for the comment. Along with a similar comment pointed out by reviewer 1, it made clear to us that we need to more precisely state the novelty and significance of our work. We have refined our language to emphasize that we identified *why and how* position has an impact and thus, revealed more predictable rules for this previously noted phenomenon. Furthermore, we more precisely discussed the novelty of our finding that TFs like NRF1 can act as direct repressors and augmented the manuscript to more broadly discuss the significance of prior work.

2. Line 56, *“Exploiting these TSSs as anchors for sequence analysis revealed that the motifs of many but not all TFs naturally displayed preferential positioning relative to sites of transcription initiation”*: This is a potential over-generalization, and we can't tell from the provided data exactly what this statement means. The authors need to clearly state in the results how many motifs for how many TFs were actually tested here (Figures and SFigures both only show possibly cherry picked examples). Otherwise, it is impossible to gauge exactly what is meant by “many but not all”; this is a qualitative statement that could at least be semi-quantitative (i.e., set a reasonable cutoff, and tell us that XX out of YY tested TFs display this behavior). Also, were these TFs chosen based on prior knowledge, or were many TFs tested in a completely unbiased fashion, and then only these few examples are shown? These important details need to be clearly indicated in the results section.

To provide more context we now provide a comprehensive analysis of TF motif enrichment based on all 463 known motifs in the HOMER database that leverages HOMER2's new background selection methods to properly assess position-specific fluctuations in nucleotide bias present near the TSS (Fig. 1b-d). Given the novelty of our findings, we initially focused our study on the most enriched TF binding sites, unbiasedly supported by de novo motif enrichment (Fig. S3a). We now have broadened our analysis across all known motifs where possible. In our biochemical analyses, we focus on NRF1, YY1, and NFY motifs due to their strong enrichment at TSS and the fact that these motifs were bound by single transcription factors (as opposed to a large family), making the factors easier to deplete and study their functional impact. The manuscript has been updated accordingly.

3. Another over-generalization, “Motif distribution patterns were more pronounced in the absence of core promoter elements (Fig. 1c, Fig. S2b).”: The statement makes it sound like it applies to all motifs, but evidence is only shown for a single TF (NFY).

We apologize that the original presentation of data was limited and as the reviewer points out, the text is over-generalized. We now include additional examples and analyses of Inr-dependent positioning (Fig. S4c-e), and have likewise narrowed our claims.

4. Line 58, “Most patterns included a 10-bp periodicity”: where is the evidence for this? Need some sort of calculation to classify which motifs are helical and which are not. There must be some simple way to gauge how close the observed periodicity is to the expected 10-11 base periodicity of DNA. The authors need to be much more systematic and quantitative to support this claim. Currently there is no evidence that I can find, even on a qualitative level.

Thank you for pointing this out. We performed fourier analysis to provide analytical support for the claim of helical binding preferences and have now included data in the new Figure S3b and Figure S4e.

5. Fig. S3b is good to see, but it is hard to fully understand the y-axis. Would these regions actually be called ChIP-seq “peaks” or are we squinting at noise? To complement the current analysis I would like to see a simple enrichment analysis where these regions are binned and the over-representation of NRF1 ChIP-seq peaks is estimated within these bins. Same thing for Fig. S5c.

To strengthen our findings we performed ChIP-seq experiments for both wild-type NRF1 and YY1 and refined our analysis. We have now added a new Figure 1f and augmented several supplemental figures to better characterize TF binding enrichment at regulated TSS (Fig. S5-S9).

6. Based on my understanding of how csRNA-seq works, the signal should be capturing multiple types of RNAs, including enhancer RNAs, pri-miRNAs, antisense transcripts, and promoter transcripts. Is there a way to estimate the relative proportion of these four “flavors” of RNAs in these analyses? TFBS location might have a different effect on eRNAs vs promoter RNAs, for example. Likewise, if eRNAs are underrepresented or have weaker/noisier signal, for example, the observed TFBS patterns could be dominated by promoters.

Thank you for this suggestion. As previously noted by Core et al 2014 (PMID: 25383968), we have observed that the TF binding site patterns in promoters and enhancers are remarkably similar. That being said, there is a clear enrichment for certain TF motifs in enhancers versus promoters (i.e. AP-1 in enhancers). To help illustrate this point, we have now added Fig. S5 which depicts the positional enrichment patterns for all 463 motifs across promoter TSS, promoter antisense TSS, and promoter-distal (i.e. ‘enhancer’) TSS. pri-miRNA TSS were omitted due to their limited number.

7. Fig. 2f is a very rich figure with lots of useful information. The underlying data need to be provided in the supplement for additional data mining purposes. For example, I would love to know what that class of TFs is that represses transcription specifically at a distance of around 50-bp upstream of the TSS.

We are glad the reviewer shares our enthusiasm for this figure. We now provide supplementary figures where all TF binding sites are labeled (Fig. S5, S12, S21) which will be provided as .pdf with sufficient resolution to read labels. We apologize that potentially due to the processing by the journal, names of our previous Fig. S10 may not have been legible.

PS: if processing may have made figures illegible they can be downloaded from https://emailwsu-my.sharepoint.com/:b:/g/personal/sascha_duttke_ws_u/EXwvmWZcgPIJoekUTPc6r-0BJEF0uwpXDyS3YIV3LvMiCA?e=VLfwef

8. Line 267, “Position-dependent interactions were observed among many motifs including CRE, Ebox, ETS, GFX, GFY, NRF1, NFY, Sp1 and YY1 but not all (i.e. LRF, FigS. 15b).”: Where are the data supporting this claim? For example, I don’t see any mention of E-boxes in Fig. 4 or Fig. S15.

We previously did not present data for all mentioned TFs in the manuscript due to space constraints. To make room for other new data in the manuscript, we have elected to retain only the relationships for TFs where we have stronger supporting data (knock-downs NRF1, NFY, and YY1) and have limited the scope of our claim (Fig. 4f,g, S20).

9. Little of no QC is provided for the functional genomics data – did the ChIP work? Did RNA-seq? Are there good QC metrics for csRNA-seq? ENCODE-type standards need to be applied and provided. For example, for ChIP-seq, I would need to see #unique mapped reads, peak counts, FRiP scores, and motif enrichment for the ChIP’d TF (relative to e.g. all motifs in the HOMER library) at minimum.

Thank you for pointing this out and we apologize for the missing QC information. We have now added ChIP-seq QC to Fig. S6b (including genome browser tracks in Fig. S6d), and have added read count information to Table S1.

10. Application of the TSS and TF motif analysis to genetic polymorphisms in humans would increase the impact of this manuscript and add a disease genetic risk component. While a comprehensive application might be outside of the scope of this manuscript, consider applying this approach to a single disease’s genetic risk variants in the context of a disease appropriate cellular context for which the lab already has csRNA-seq.

We appreciate the exciting suggestion! We agree and took advantage of published TSS data from 67 individuals of the Yoruba population [PMID: 33235186] to define variant-TSS relationships (tssQTLs). The tssQTLs can be analyzed using the same new approaches developed in HOMER2 to rigorously assess the position-dependent impact of variants found in TF binding sites (Fig. 5a-c). We demonstrate how this data largely supports the same conclusions drawn from other data in the manuscript, and highlight that tssQTLs associated with disease-risk variants defined by GWAS can have differential impact on initiation depending on their relative position to the TSS (Fig 5b,c).

Minor comments

1. Please provide the quantified level of TF knockdown for each siRNA in Table S1.

Done - provided in Fig. S6a.

2. Refs 1 and 2 do not appear to support the statement they are meant to support (one is on flower development in plants and the other involves fly segment development). These are likely mistakes. Ref 3 makes sense at least.

Thank you for the comment. The genes defined in Ref 1&2 to be responsible for shaping cellular and morphological diversity were later found to be TFs. But we agree that more appropriate citations may include:

Kasowski, M. *et al.* Variation in transcription factor binding among humans. *Science* **9**;328(5975):232 (2010)

Takahashi, K. & Yamanaka, S. Induction of pluripotent stem cells from mouse embryonic and adult fibroblast cultures by defined factors. *Cell* **126**, 663–676 (2006).

We have updated the manuscript accordingly.

3. Line 40, typo: 'TF motifs pacing'.

Corrected - thank you for catching that!

4. Fig. S1c is never really described in the text. I can't tell what it is showing. What exactly is MEIRLOP doing?

We apologize that we were not clear with this analysis and have adjusted the text. MERILOP is a motif enrichment analysis software that scores the presence of motifs in a set of *ranked* input sequences. In this case, we examined which motifs were most strongly associated with TSS strength (i.e. csRNA-seq read count). As such, it does not just define motifs enriched in the input sequences versus a background, but scores motifs based on their enrichment or depletion in sequences with higher ranking (i.e. higher TSS activity). The new legend to Fig. 1c is:

"Motifs enriched in regulatory regions with TSS supporting weak or strong transcription initiation. Motif Enrichment In Ranked Lists of Peaks (MEIRLOP) (Delos Santos et al. 2020) was used to determine enrichment of motifs in U2OS regulatory regions scored by TSS strength, while controlling for sequence bias."

5. Fig. S1f is also never described in the text, but the results look interesting and important.

Yes, thank you for pointing this out. Fig S1f shows that on average, TF binding sites located in near -30 and -50 bp, relative to the TSS, are more likely conserved across species than elsewhere, thereby providing correlative evidence for the importance of motif spacing to the TSS. We had trouble working this into the manuscript given the length restriction and flow, but we have added *"Positional preferences were conserved across cell lines, vertebrate species, and TSS detection methods, and in some cases were restricted to genomic locations with cell type-specific activity (e.g., HNF1, Fig. S3d-f)."*

6. Official gene names should be included where possible (for example, LRF=ZBTB7A).

We are happy to change this. We changed LRF/ZBTB7A to ZBTB7A/LRF and used ZBTB7A in figures now. We originally used LRF due to its prevalent usage in the literature, but agree it is important to use the official gene names whenever possible to avoid confusion.

7. Line 104: I think the figure refs should be Fig. 1b,d (and not c,d).

Thank you for pointing this out. We agree and we have edited the manuscript.

8. Calling YY1 an activator is misleading; its name (Ying Yang 1) is based on the fact that it can activate or repress transcription.

We agree, have refined the manuscript, and apologize for the inaccuracy.

9. Fig. 2: It would be a lot less confusing if the red "activates" label was on top and the blue "represses" label was on the bottom (currently they are in the opposite order – the colors match up but not their positions). This is a point we had intensively discussed among ourselves when making the figure. As it is, red labels regions where the *motif activates under wild type conditions*, blue where it represses. Of course, this figure shows mutations. Perhaps as we admire geneticists for their ability to think "backwards", we found inverting the colors helpful. We felt this presentation was more intuitive and in line with the general color scheme of the manuscript of red activating transcription, blue decreasing it.

10. Line 154: Should be (Fig. 2f).

Thank you for this comment. Fig. 2e is leading into Fig. 2f by showing an example of each using the same plots as in c,d (the idea being from 1 to 3 to all). But we agree that the potentially better referral would be to both Fig. 2e,f.

11. Citation on line 182 is messed up.

Corrected!

12. Fig. 3c might be clearer if the y-axes had identical ranges for all of the panels.

Unfortunately, as CTCF goes to -8, this would make both YY1 and p53 much harder to read. We hope the inclusion of the red line at y=0 helps with the interpretation of the data.

13. The x-axis label in Fig. 4a is misleading; I think it should range from -100 to +40. Same with S12 all panels.

Good suggestion. The old panels described the “DNA insert” in the analysis which, we agree, could be simplified to spacing relative to the TSS.

14. Fig. S7 is very hard to read.

We agree. We did our best to correct it (copied a 600 dpi picture in) and attached the .ai file to the reviews.

Upon publication, this figure files as .pdf will also be made available for download which allows complete zoom in (i.e. 300 dpi)

15. The FAQ for TSS-MPRA is potentially useful. Suggestions below:

a. The FAQ currently reads more as a very helpful blog than manuscript methods protocol. In multiple sections, the FAQ shares the success or lack of success for specific approaches – this is likely to change over time. I would suggest instead posting this online and providing a url in the manuscript to provide an opportunity to update the protocols/FAQ as additional considerations and optimizations are available.

b. If this will be part of the manuscript, more details are needed. For example, “ensure you have a good transfection rate” needs to include the transfection rate that would be considered “good”. Under “how much RNA can we use as input?” be specific about the higher ratio increasing or decreasing sensitivity – rather than using the non-directional “influence”.

c. Under “how diverse can my plasmid library be?”, consider defining diversity and complexity. Instead of presenting a cost-based argument, consider simply providing the amount of sequencing needed and allowing readers to make their own cost-based assessments.

d. It was unclear what “nascent style” referred to in the section “Can TSS-MPRAs be performed nascent style?”.

e. Many abbreviations are included without definition. While most of these are common experimental phrases, definitions are needed for clarity.

Thank you for the positive feedback and helpful suggestions. For csRNA-seq we published the FAQ but are maintaining an up-to-date version on our HOMER website

<http://homer.ucsd.edu/homer/ngs/csRNAseq/index.html>. We have revised the FAQ (posted below) but following your suggestion, we have removed this part from the manuscript and are in process of adding the tutorial to the HOMER website. Revisions of the FAQ are as follows:

TSS-MPRA troubleshooting and FAQ:

The following FAQ provides answers to commonly asked questions encountered when performing TSS-MPRA experiments, and is included to improve reproducibility. Please note that all the following answers were offered to the best of our *current* knowledge.

- *I failed to get a library. Where to start troubleshooting?*
 - Ensure you are not losing your RNA (i.e. using Qubit) - The quantity of RNA should not significantly change (< 10%) during the assay.
 - Given that the assay uses a lot of RNA as input, unexpected RNase contamination is unlikely to be an issue. Yet, it would be wise to briefly test your reagents for RNase activity i.e. using IDT's RNase Alert or similar.
 - Ensure that you have a good (>80 %) transfection rate (i.e. a plasmid with GFP).
 - Include a strong promoter such as CMV or the SuperCorePromoter (SCP) used in the HARPE assay ⁴⁹ as a true positive.
 - Generating a stable transcript makes the assay easier. Consider cloning your regulatory elements of interest in front of a stable ORF (i.e. GFP). Once you use a stable reporter mRNA, you could also consider increasing the duration between transfection and harvest. This will increase the % of desired product in the final mixture but comes with its own caveats (see below). Hence this is rather a suggestion for troubleshooting more than a standard way of running the assay.

- *How diverse can my plasmid library be?*

In this manuscript we utilized pools with 2000 oligos but we have successfully generated TSS-MPRA libraries from plasmid pools with much higher diversity (250-1M unique sequences). Your limitation will likely lie in the detection on the RNA level - problems can arise when a few strong promoters attribute most of the signal, which can make getting high coverage of complex libraries costly. As such, to interrogate highly complex libraries, it may be best to generate multiple pools, transfect them independently, and then do multiple TSS-MPRAs or strategically pool RNA from multiple transfections as input for one assay.

- *How much RNA can we use as input? Does more RNA help?*

We successfully generated libraries with anywhere between 3-30 µg of total RNA but recommend 10 - 15 µg to start with. We do not recommend using more than 30 µg RNA. The ratio of RNA transcribed from your plasmid to what's transcribed in the genome influences the assay's sensitivity. Therefore, the assay is better optimized by increasing reporter activity, rather than by adding more RNA. as all these transcripts compete for the 5' adapter.

- *How long is it best to wait prior to extracting the RNA following transfection?*

As short as possible to avoid the impact of differential transcript stability, but longer times help with detection by allowing the plasmid templated transcripts to accumulate. We recommend starting with ~8 hours when using Lipofectamine (in our hands it took about 7 hours to start seeing decent GFP signal) and less when using electroporation. While longer incubation times help with detection and may be a good way to troubleshoot your experiment, 5' ends of mRNAs can get post-transcriptionally modified ⁶¹ which will generate noise in your data. In this sense, if possible, it should be better to let your sequences initiate an unstable RNA without splice sites and open reading frames (ORFs).

- *Can TSS-MPRA be performed to capture nascent transcription?*

Yes, plasmids are in the nucleus so you can purify nuclei and then perform a 5' run-on method such as PRO-cap or 5'GRO-seq. This experiment, however, is technically very challenging and at least in our hands, no significant differences were observed when the RNA was harvested within 10 hours following the Lipofectamine transfection.

- *Can I add RNase after reverse transcription but before PCR?*

Yes. In our hands, however, the assay worked better when RNase was added after the PCR.

- *I just want to quickly screen a ton of constructs... How can I speed this assay up?*

You do not have to purify the RNA following the 5' adapter ligation to do the annealing. Doing so, however, improves the quality and robustness of the assay.

- *Can I use SPRI beads instead of Trizol for RNA purification when doing TSS-MPRA?*

This is tricky. QuickCIP [NEB] is very stable and beads carry over protein. We therefore do not advise it.

We thank the reviewer for the thoughts and suggestions.

Reviewer Reports on the First Revision:

Referees' comments:

Referee #1 (Remarks to the Author):

The authors have fully addressed my comments. I am happy to support publication of this paper. Congratulations to the authors on an outstanding contribution.

Charles Danko

Referee #2 (Remarks to the Author):

The authors have responded in a satisfactory manner to my previous criticisms. I have no further comments.

Referee #3 (Remarks to the Author):

The authors have done an excellent job of responding to my comments. I am happy with this revised version.

Matt Weirauch (with Leah Kottyan)

Referee #3 (Remarks on code availability):

This group has a very strong track record of producing high quality, well-documented code. I suspect this current code is no different.

I have not reviewed the code due to current time constraints on my schedule. I would be happy to do so if needed, if I had another week or two.